# High Performance Space Debris Tracking in Complex Skylight Backgrounds with a Large-Scale Dataset

## Abstract

With the rapid development of space exploration, space debris has attracted more attention due to its potential extreme threat, leading to the need for real-time and accurate debris tracking. However, existing methods are mainly based on traditional signal processing, which cannot effectively process the complex background and dense space debris. In this paper, we propose a deep learning-based Space Debris Tracking Network (SDT-Net) to achieve highly accurate debris tracking. SDT-Net effectively represents the feature of debris, enhancing the efficiency and stability of end-to-end model learning. To train and evaluate this model effectively, we also produce a large-scale dataset Space Debris Tracking Dataset (SDTD) by a novel observation-based data simulation scheme. SDTD contains 18,040 video sequences with a total of 62,562 frames and covers 250,000 synthetic space debris. Extensive experiments validate the effectiveness of our model and the challenging of our dataset. Furthermore, we test our model on real data from the Antarctic Station, achieving a MOTA score of 73.2%, which demonstrates its strong transferability to real-world scenarios. Our dataset and code will be released soon.

## 1 Introduction

During the past decade, global space activities have exponentially increased, which has raised the potential risk of collisions between debris and spacecraft in near-Earth space Katz (2024); Steindorfer et al. (2025). This has led to significant economic losses, drawing increasing attention to the issue of space debris. The Space Debris Tracking (SDT) is to detect and predict the trajectories of space debris, reducing the collision risks and promoting space industry development. This makes monitoring debris extremely urgent and critical for ensuring the safety of space operations.

SDT is an essential task that includes detecting and predicting the trajectories of space debris. It is a computer vision problem, specifically an object tracking task in astronomical videos, where the goal is to identify and follow space debris across continuous frames for a video. However, traditional methods typically focus on object debris detection rather than tracking. These approaches often rely on techniques such as template-matching methods Liu et al. (2012); Murphy et al. (2017) and morphological operator methods Wei et al. (2018); Jiang et al. (2022b). While these methods are capable of detecting debris in individual images, they struggle to associate objects across consecutive frames, making them less suitable for space debris monitoring.

Artificial intelligence (AI) has achieved significant success in fields such as object detection Zou et al. (2023) and object tracking Javed et al. (2022), due to its exceptional ability to process large-scale datasets. Traditional methods have limitations in space debris detection and cannot effectively handle long-term tracking tasks. AI models can make up for this deficiency through efficient data processing and learning methods. However, AI models rely on large-scale, high-quality datasets while astronomy data are extremely limited. So, there is no an effective AI model that could meet the requirements of debris tracking mission for popular telescopes like the Legacy Survey of Space and Time (LSST) Tucker (2023) and Wide-Field Survey Telescope (WFST) Wang et al. (2023).

To address the challenge of limited data in astronomy, we follow the idea of observation-based data simulation in the astronomical field Lin et al. (2021); Peterson et al. (2015); Perrin et al. (2014). The

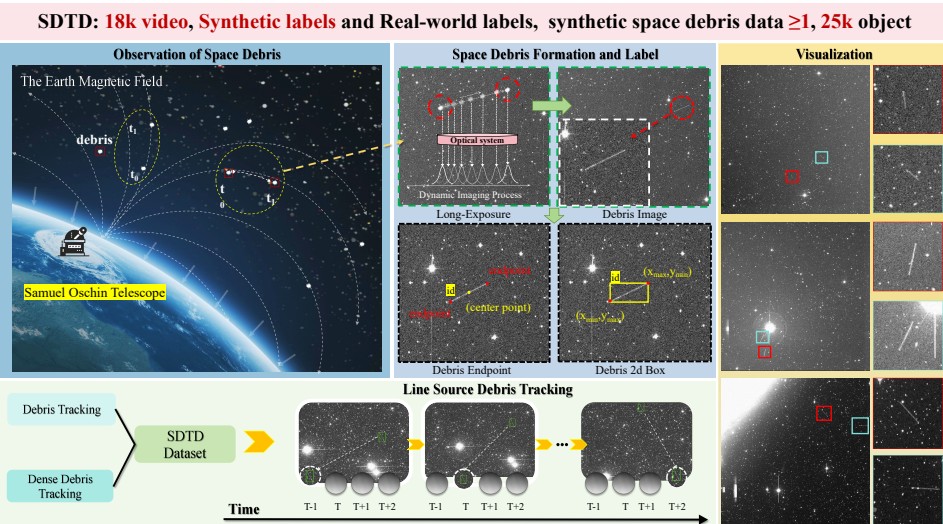

Figure 1: Illustration of the space debris formation and tracking pipeline using synthetic labels. It includes the simulation of space debris within Earth's magnetic field, the generation of synthetic labels such as endpoints, center points, and bounding boxes for debris in image frames. Additionally, the process of line source debris tracking across multiple frames is shown, with visualization examples of debris tracking.

observation-based data simulation is widely proven to lead to realistic data and reliable scientific conclusions, providing a chance for us to design a method for data simulating space debris. Due to its success, we construct a Space Debris Tracking Dataset (SDTD), which is the first publicly available dataset designed for space debris tracking. Fig 1 shows the formation principle, type, examples, and corresponding SDT tasks of space debris. SDTD consists of 18,040 synthetic videos, totaling 62,562 frames and 25,000 annotated debris instances.

In order to apply AI methods to the SDT task, we propose a simple yet effective deep learning method, namely the Space Debris Tracking Network (SDT-Net). We introduce a Region-of-Interest Feature Enhancement (RoIFE) module to highlight debris features, aiding in more accurate detection. A detection module is then used to localize the debris by predicting their positions in the image. After localization, a debris offset module is applied to perform cross-frame data association, generating tracking trajectories for each debris. In summary, the key contributions of this paper are as follows:

1. We propose a space debris simulation method. Based on this method, we construct SDTD, the first benchmark dataset for SDT. It contains 18,040 video data with complex backgrounds and diverse debris targets, and the data is close to the real space debris situation.

2. We introduce the Space Debris Tracking Network (SDT-Net) to improve space debris tracking. By integrating a RoIFE module, a detection module, and a tracking module, we effectively enhance debris representations, leading to more accurate and reliable tracking performance.

3. Evaluation on the constructed SDTD dataset shows that the proposed SDT-Net achieves state-of-the-art performance in terms of space debris tracking on the test set overall.

## 2 RELATED WORK

### 2.1 SPACE DEBRIS DETECTION AND TRACKING

Space debris detection relies on the precise localization of targets within optical imagery. Such as Jiang et al. Jiang et al. (2022a), utilized traditional image processing techniques, employing enhanced median filtering and improved Hough transforms to isolate targets from noise. Guo et al. Guo et al. (2025) proposed an enhanced YOLOv8-based architecture, achieving notable improvements

in both detection accuracy and inference speed. Similarly, Jia et al. Jia et al. (2020) developed a deep neural network framework specifically addressing low signal-to-noise targets by integrating Faster R-CNN Ren et al. (2015) with a modified ResNet-50 He et al. (2016) backbone. For tracking, Cament et al. Cament et al. (2021) introduced a Poisson Labeled Multi-Bernoulli multi-target tracking filter that incorporates probabilistic modeling and orbital-dynamics constraints. However, these related studies focus on satellite Zhou et al. (2023); Roll et al. (2023) and point debris targets Guan et al. (2025); Tao et al. (2023), but these methods are based on the assumption of point source detection or tracking, which makes them unable to generalize to the more complex debris appearances found in real-world observation data. In this paper, we extend the detection to complex tracking, playing a key role in advancing space exploration.

## 3 BENCHMARK DATASET

In this paper, we utilize astronomical images collected by the Zwicky Transient Facility (ZTF) Bellm (2014) to simulate space debris. We propose a simulation tool to generate space debris, which provides sufficient data for constructing the SDTD dataset. Under long-exposure conditions in telescopic observations, debris often appears as a line source in the images. So, our simulation target is to generate realistic long-exposure line sources. We develop a pipeline that includes data pre-processing, line source debris video generation, and post processing. Fig 2 and the following paragraphs provide a detailed description of this process.

### 3.1 DATA COLLECTION AND PRE-PROCESSING

#### 3.1.1 DATA COLLECTION

The Zwicky Transient Facility (ZTF) is a ground-based optical survey project designed to monitor astronomical transient phenomena. It operates on the 48-inch Samuel Oschin Telescope at Palomar Observatory, using a 47-square-degree, 600-megapixel CCD camera for high-cadence, wide-field imaging. Its capability to capture large-scale, high-resolution astronomical images makes it suitable for simulating space debris.

In order to obtain enough data for simulation, we collected 17,235 ZTF astronomical observation data from June 2018 to June 2022. These data contain complex skylight background information, including atmospheric scattered light, moonlight, and human factors such as light pollution. Based on these richer astronomical observation data, we conduct simulations and generated debris videos.

Some regions of the images contain outliers, appearing as zeros or extremely high values, which leads to numerical overflow issues. To ensure the quality of our dataset, we carefully examine the dataset, remove 1,195 unusable images, and retain 16,040 high-quality images.

#### 3.1.2 DATA PRE-PROCESSING

Unlike natural images, astronomical images typically contain larger pixel values. Therefore, a data processing step is necessary to transform these values into the standard range of grayscale image pixels. To achieve this, we apply the ZScale method Payne et al. (2003) to map image pixel values to a reasonable range. The ZScale method first calculates the median $z_{median}$ and standard deviation $z_{std}$ of the image pixels. Then uses these two values to estimate a range $[z_1, z_2]$:

$$z_1 = z_{median} - k \times z_{std}, z_2 = z_{median} + k \times z_{std}, \quad (1)$$

where $z_1$ and $z_2$ represent the lower and upper bounds of the pixel intensity range, respectively. The hyperparameter $k$ is assigned a value of 2.5. Then, we perform a linear stretch on the grayscale values of the image, mapping the range $[z_1, z_2]$ to $[0, 255]$:

$$I_{scaled}(x, y) = 255 \times \frac{I(x, y) - z_1}{z_2 - z_1}. \quad (2)$$

Finally, pixels with grayscale values below $z_1$ or above $z_2$ are set to 0 or 255:

$$I_{\text{final}}(x, y) = \begin{cases} \max(I_{\text{scale}}(x, y), 0), & I_{\text{scale}}(x, y) < 0 \\ \min(I_{\text{scale}}(x, y), 255), & I_{\text{scale}}(x, y) > 0. \end{cases} \quad (3)$$

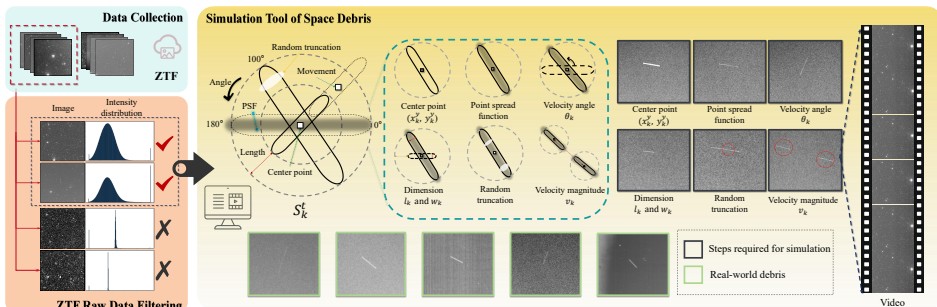

Figure 2: Overview of the space debris simulation pipeline. Step 1: Data Collection. Step 2: ZTF Raw Data Filtering. Step 3: Simulation Tool for Space Debris. Various parameters are applied to simulate space debris. The process generates synthetic video sequences and shows the real-world debris for reference.

By enhancing the contrast and detail of astronomical images, the ZScale method ensures a reliable basis for the next simulations. After pre-processing the data, we conduct simulations to generate space debris, providing sufficient data for constructing the SDTD dataset.

## 3.2 SIMULATION OF SPACE DEBRIS

For a single astronomical image, the Space Debris Simulation generates a video with $T$ frames of line source debris, simulating the movement of debris across each frame.

### 3.2.1 LINE SOURCE DEBRIS VIDEO GENERATION

The goal of the space debris simulation is to generate a sequence video with multiple line source debris. For each debris (set $k$-th as an example), we represent it as a parameter sequence $\{S_k^1, S_k^2, S_k^3, ...S_k^T\}$ where $T$ is the total frame number. $S_k^t$ is the debris parameters for the $t$-th frame, which is set as follows:

$$S_k^t = \{\underbrace{x_k^t, y_k^t}_{\text{center}}, \underbrace{l_k, w_k}_{\text{dimension}}, \underbrace{\theta_k, v_k}_{\text{velocity}}\}, \tag{4}$$

where $(x_k^t, y_k^t)$ represents center point coordinates of the line source at time $t$. $l_k, w_k, \theta_k$ and $v_k$ are line length, line width, velocity angle and velocity magnitude, respectively.

With this parametrized line source, we randomly sample a value $K$ as the total line source number and then for the $k$-th line source, we first generate its $S_k^1$ by randomly sampling its 6 parameters. The generated $S_k^1$ provide the initial center point locations, defined dimension, and a fixed velocity of the debris. So, we utilize these parameters to produce subsequent parameters $S_k^2 \sim S_k^T$.

Take the second frame ($t = 2$) as an example, we calculate the parameters for $S_k^2$. The center point coordinates $(x_k^2, y_k^2)$ is calculated by combining velocity and initial center as follows:

$$\begin{aligned} x_k^2 &= x_l^1 + v_k \cdot \cos(\theta_k) \cdot \Delta t, \\ y_k^2 &= y_k^1 + v_k \cdot \sin(\theta_k) \cdot \Delta t, \end{aligned} \tag{5}$$

where $\Delta t$ represents the time interval between consecutive frames, which is set to 1 here for simplification. Thus, the parameters for the second frame are represented as $S_k^2 = \{x_k^2, y_k^2, l_k, w_k, \theta_k, v_k\}$. The same procedure is repeated for each subsequent frame $t = 3, 4, 5, ..., T$. This process continues frame by frame, until all parameters $\{S_k^1, S_k^2, S_k^3, ...S_k^T\}$ are generated.

We proceed to generate the corresponding line source in the image. For each frame $t$, we take the parameters $S_k^t = \{x_k^t, y_k^t, l_k, w_k, \theta_k, v_k\}$ and follow three steps to generate the line source. With these, at the center point coordinates $(x_k^t, y_k^t)$, we draw a rectangle with dimensions, $l_k$ and $w_k$, and angle $\theta_k$. Then, we apply random brightness noise to the rectangle. By traversing $k$, we draw entire $K$ objects and if the rectangle is out of the image, we would neglect it.

Figure 3: The overall pipeline of our SDT-Net. The inputs are the current frame $I_t$, the previous frame $I_{t-1}$ and heatmap $H_{t-1}$. The outputs include amodal endpoints heatmap, line source embedding, and tracking offset.

At last, using the parameters $\{S_k^1, S_k^2, S_k^3, ...S_k^T\}$, we generate the corresponding images $\{I^1, I^2, I^3, ..., I^T\}$, resulting in a video that captures the movement trajectory of the debris.

### 3.2.2 POST PROCESSING FOR REALISTIC

Since the real line source debris conforms to the Point Spread Function (PSF) characteristics Koupri-anov (2008), we perform point diffusion function processing on the preliminary line source debris. Among them, PSF is expressed as follows:

$$I_k^t(x, y) = \frac{S}{2\pi\delta_{psf}^2} exp(-\frac{(x - x_c)^2 + (y - y_c^2)}{2\delta_{psf}^2}), \qquad (6)$$

where $I_k^t(x, y)$ is the pixel value of the line source debris, $(x_c, y_c)$ represents the center point coordinates of the line source debris, $S$ is the scale factor, and $\delta_{psf}$ is the diffusion standard deviation of the imaging system PSF. Additionally, we apply random truncation to the line source to simulate more realistic conditions. This operation enables us to simulate realistic line source debris.

### 3.3 DATASET STATISTICS

After the simulation, we obtained 18,040 videos with a total of 62,562 frames. To evaluate the performance of the model in complex environments, we randomly selected 1,000 images from the 16,040 original data and constructed two test sets: the fragmented test set and the dense fragmented test set. The debris test set contains 1 to 2 debris, and the dense debris test set contains 1 to 5 debris. For more details and statistics of the dataset, please see the supplementary material.

## 4 METHOD

### 4.1 OVERVIEW

The entire pipeline of our Space Debris Tracking Network (SDT-Net) method is shown in Fig. 3, which is a CenterTrack-style Zhou et al. (2020) tracking method. Due to the debris long-exposure as linear, we treat the debris tracking as tracking for linear segments. For each object, we follow the tracking-by-detection idea. Our model detects two endpoints of each line and predicts velocity for cross-frame object association to generate trajectory.

Specifically, a backbone extract base features for each frame, we establish a Region-of-Interest Feature Enhancement (RoI-FE) module to highlight the debris object cues in feature maps. Then, a novel design detection module consists of an endpoint heatmap head and a line source embedding head that collaborate to localize two endpoints of each debris. After that, a tracking head predicts velocity for each object by computing endpoint offsets across frames.

## 4.2 BACKBONE

For the $t$-th frame, the model takes three components as inputs, the image of the current frame, the previous frame and its predicted endpoints representing as heatmaps. Each input is passed through a corresponding input-dependent module containing a convolutional layer, a batch normalization layer, and a ReLU activation function. After this, the three feature maps are added together, resulting in a unified feature map $F_{input} \in \mathcal{R}^{W_{in} \times H_{in} \times 3}$. The feature map $F_{input}$ is then passed through our backbone, a DLA model Yu et al. (2018), which consists of an encoder and a decoder. The encoder outputs a series of feature maps at different resolutions, denoted as $\{F_1, F_2, F_3, F_4\}$. The decoder, an IDA up-sampler Yu et al. (2018), takes these feature maps to gradually generate upsampled feature maps and we take the highest resolution one as our backbone feature $F_b \in \mathcal{R}^{W \times H \times C}$.

## 4.3 REGION-OF-INTEREST FEATURE ENHANCEMENT

The Region-of-Interest Feature Enhancement module first computes a debris mask. Specifically, we introduce another IDA up-sampler, sharing the same structure with the backbone decoder but having independent parameters, as our segmentation head. Its output channels are set to 1 to obtain a debris mask $\hat{M} \in \mathcal{R}^{W \times H \times 1}$ based on encoder features. The output mask is multiplied by $F_b$ as follows:

$$F_{\text{en}} = \hat{M} \odot F_b \tag{7}$$

to acquire the enhanced feature $F_{en} \in \mathcal{R}^{W \times H \times C}$. $F_{en}$ highlights debris features and removes background noise. To ensure that the obtained mask $\hat{M}$ accurately indicates the desired debris region, we use a Segmentation Loss to train the segmentation head. The segmentation mask $M$ are generated by connecting the ground truth endpoints to form a segment and is supervised by the ground truth mask using the following loss function:

$$\mathcal{L}_{\text{seg}} = -\frac{1}{WH} \sum_{i=1,j=1}^{W,H} \left[ M_{ij} \log(\hat{M}_{ij}) + (1 - M_{ij}) \log(1 - \hat{M}_{ij}) \right], \tag{8}$$

where $W, H$ are image dimensions, $M_{ij}$ is the ground truth, and $\hat{M}_{ij}$ is the predicted probability. The obtained $F_{en}$ is then passed through two modules: a Line Source Detection module and a Debris Offset module.

## 4.4 LINE SOURCE DETECTION MODULE

In each frame, we detect each line source by detecting its two endpoints. Two endpoints of each line source must be one on the left and one on the right. So, we design our model to produce a left-endpoint heatmap and a right-endpoint heatmap. The left-endpoint heatmap indicates left endpoints of all line sources, where the right one reflects right points. Then, we establish an endpoint embedding module to match different left and right endpoints belonging to a same line source together.

### 4.4.1 ENDPOINTS HEATMAP

The endpoints heatmap head is a convolutional layer that takes $F_{en}$ as input and feedback heatmaps $\hat{H} \in \mathcal{R}^{W \times H \times 2}$, which has two channels that one for the left and one for the right endpoint. Each pixel in the heatmap represents the likelihood of the position being a line source endpoint.

Follow existing detection heatmap training pipeline, we utilize the Focal Loss Lin et al. (2017) to train the heatmap head as follows:

$$\mathcal{L}_{\text{hm}} = \frac{1}{N_{\text{pos}}} \sum_{whc} \text{Focal}(H, \hat{H}), \tag{9}$$

where $\hat{H}$ and $H$ are the predicted and the ground-truth heatmaps. $w, h, c$ means the spatial and channel index. The ground-truth heatmaps $H$ are generated by drawing Gaussian kernels at the positions of each endpoint.

### 4.4.2 LINE SOURCE EMBEDDING

The line source embedding predicts features for each endpoint. Specifically, we predict a left endpoint line source embedding maps $\hat{E}_l \in \mathcal{R}^{W \times H \times C_2}$ and a right endpoint line source embedding maps $\hat{E}_r \in \mathcal{R}^{W \times H \times C_2}$. During inference, with the predicted endpoints heatmaps, we first derive local-maximum points for each heatmap, resulting in $N$ left endpoints and $M$ right endpoints. Then we utilize there points to take corresponding embeddings from $\hat{E}_l$ and $\hat{E}_r$, respectively, denoted as $\{e_l^1, e_l^2, ..., e_l^N\}$ and $\{e_r^1, e_r^2, ..., e_r^M\}$. Based on these, we would compute an embedding similarity matrix $S^{emb} \in \mathcal{R}^{N \times M}$ where $S_{i,j}^{emb}$ is computed as follows:

$$S_{i,j}^{emb} = |(e)_l^i - (e)_r^j|. \tag{10}$$

$S_{i,j}^{emb}$ reflects similarity of $i$-th left point and $j$-th right point. With this, we provide a simple yet effective scheme, that each left point matches with its nearest neighbors right point to organize an endpoint pair as a detected object.

Inspired by CornerNet Law & Deng (2018), we design the loss function of our embedding as follows:

$$\mathcal{L}_{\text{same}} = \frac{1}{K} \sum_{k=1}^{K} \left[ (e_l^k - e_c^k)^2 + (e_r^k - e_c^k)^2 \right], \tag{11}$$

where $e_l^k$ and $e_r^k$ are endpoint features, $e_c^k$ is the mean of $e_l^k$ and $e_r^k$, utilized to push $e_l^k$ and $e_r^k$ as close as possible. Meanwhile, we introduce $\mathcal{L}_{\text{diff}}$ to separate the features of different pairs as:

$$\mathcal{L}_{\text{diff}} = \frac{1}{K(K-1)} \sum_{k=1}^{K} \sum_{\substack{j=1 \\ j \neq k}}^{K} \max \left( 0, \Delta - |e^k - e^j| \right), \tag{12}$$

where $\Delta$ is the minimum feature distance threshold. In our experiments, we set $\Delta$ to 1. This term facilities to push center features belonging to different objects far away.

### 4.5 DEBRIS OFFSET MODULE

We construct a Debris Offset module. For the $t$-th frame, this module predicts two offset maps, $\hat{O}_l^t \in \mathcal{R}^{W \times H \times 2}$ and $\hat{O}_r^t \in \mathcal{R}^{W \times H \times 2}$ for left and right endpoints, respectively.

During inference, the offset plays a key role in data association. Assume that the detection module provides $N$ detected objects, we could index $N$ offsets as: $\{(o_l^t)_1, (o_l^t)_2, (o_l^t)_2..., (o_l^t)_N\}$ and $\{(o_r^t)_1, (o_r^t)_2, (o_r^t)_2..., (o_r^t)_N\}$. They indicate object endpoint offsets between the current frame and the previous $t-1$ frame. With these predicted offsets, for each detected object, we could derive endpoint locations at the previous frame as follows:

$$\text{left: } (c')_l^{t-1} = c_l^t - o_l^t, \text{ right: } (c')_r^{t-1} = c_r^t - o_r^t, \tag{13}$$

where $c'$ is the derived endpoint coordinate and $c$ is the detected one. These derived endpoints imply the potential previous object locations. With these, we could match objects across frames by measuring the location consistency between the derived endpoints and the detected object endpoints. If the previous frame has $M$ detected objects, we would compute a similarity matrix $S^{obj} \in \mathcal{R}^{N \times M}$ where $S_{i,j}^{obj}$ is computed as follows:

$$S_{i,j}^{obj} = |(c')_l^{t-1} - (c)_l^{t-1}| + |(c')_r^{t-1} - (c)_r^{t-1}|. \tag{14}$$

$S_{i,j}^{obj}$ reflects similarity of $i$-th object in the $t$-th frame and $j$-th object in the $t-1$ frame. Similar to the embedding module, we also utilize nearest neighbor as a matching scheme that the object in the current frame associates with its nearest neighbors in the previous frame.

To achieve the aforementioned function, during the training phase, we minimize the difference between the ground-truth offset and the predicted offset as follows:

$$\mathcal{L}_{\text{off}} = \frac{1}{K} \sum_{i=1}^{K} \left| (o^t)_i - [(c^t)_i - (c^{t-1})_i] \right|. \tag{15}$$

### 4.6 LOSS FUNCTION

Finally, the loss function $\mathcal{L}$ is defined by the weighted summation of all loss terms,

$$\mathcal{L} = \lambda_{\text{seg}}\mathcal{L}_{\text{seg}} + \lambda_{\text{hm}}\mathcal{L}_{\text{hm}} + \lambda_{\text{emb}}(\mathcal{L}_{\text{same}} + \mathcal{L}_{\text{diff}}) + \lambda_{\text{off}}\mathcal{L}_{\text{off}}, \tag{16}$$

where $\lambda_{\text{seg}}$, $\lambda_{\text{hm}}$, $\lambda_{\text{emb}}$ and $\lambda_{\text{off}}$ represent the weights for $\mathcal{L}_{\text{seg}}$, $\mathcal{L}_{\text{hm}}$, $(\mathcal{L}_{\text{same}} + \mathcal{L}_{\text{diff}})$ and $\mathcal{L}_{\text{off}}$.

## 5 EXPERIMENTS

### 5.1 EXPERIMENTAL SETUP

All experiments are performed using the mmdetection framework Chen et al. (2019) with Pytorch. The SDT-Net model is trained on 8 NVIDIA 4090 GPUs. For more details of the experimental, please see the supplementary material. The entire model is trained for 60 epochs, and the initial learning rate is set to 3e-3, which is decayed by 0.1 at the 20 epoch. We employ the resolution of the input images is set to 1524×1524 and the batch size is set to 2. Besides, the value of $\lambda_{\text{seg}}$ and $\lambda_{\text{emb}}$ are set to 1.0, $\lambda_{\text{hm}}$ is set to 10, and $\lambda_{\text{off}}$ is set to 0.1.

Table 1: Comparison of trackers on Debris Test and Dense Debris Test. The table compares the performance of various tracking algorithms in two distinct scenarios. The **best** results are highlighted in **red**, and the **second-best** are in **blue**. $n$ represents the number of debris.

| Tracker | Debris Test ($n \leq 2$) | | | | | | Dense Debris Test ($1 \leq n \leq 5$) | | | | | |
|---|---|---|---|---|---|---|---|---|---|---|---|---|
| | IDF1↑ | MOTA↑ | IDS↓ | HOTA↑ | DetA↑ | AssA↑ | IDF1↑ | MOTA↑ | IDS↓ | HOTA↑ | DetA↑ | AssA↑ |
| LSD Von Gioi et al. (2012) | 11.4 | 9.8 | / | / | / | / | / | / | / | / | / | / |
| Deepsort Bewley et al. (2016) | 71.7 | 55.3 | 211 | 56.4 | 56.9 | 55.8 | 72.9 | 57.8 | 1947 | 58.7 | 60.1 | 57.4 |
| Motdt Chen et al. (2018) | 69.1 | 54.0 | 239 | 54.2 | 55.6 | 52.8 | 68.6 | 53.5 | 2015 | 53.9 | 55.7 | 52.2 |
| CTracker Peng et al. (2020) | 83.4 | 71.0 | 240 | 72.9 | 74.3 | 71.5 | 69.3 | 61.7 | 1660 | 58.2 | 63.9 | 53.0 |
| Sort+YoloX Ge (2021) | 85.6 | 74.7 | **130** | 75.1 | 75.3 | 74.8 | 76.0 | 59.3 | 1394 | 61.5 | 61.7 | 61.3 |
| CenterTrack Zhou et al. (2020) | 86.3 | 77.1 | 174 | 81.0 | **85.6** | 75.9 | 72.5 | 64.8 | 1292 | 66.1 | **76.7** | 56.9 |
| ByteTrack Zhang et al. (2022) | 91.0 | 83.3 | 195 | 82.8 | 83.9 | 81.6 | 79.4 | 62.8 | **1211** | 66.3 | 66.8 | **65.8** |
| OCSORT Cao et al. (2023) | **91.6** | **84.1** | **138** | **83.7** | 84.9 | **82.5** | **82.6** | **68.5** | 1260 | **70.8** | 73.3 | **68.3** |
| SDTNet (Ours) | **91.8** | **87.7** | 169 | **87.8** | **90.9** | **84.8** | **80.6** | **70.3** | **1070** | **73.6** | **81.6** | 65.6 |

### 5.2 EVALUATION METRICS

We use standard MOT Bernardin Keni (2008) metrics, including MOTA, IDF1, and IDs, which provide a comprehensive assessment of tracking performance. We also use HOTA Luiten et al. (2021) metrics. Notably, HOTA combines Detection Accuracy (DetA), which measures detection precision, and Association Accuracy (AssA), which evaluates identity consistency across frames.

### 5.3 COMPARISON WITH EXISTING MOT METHODS

As shown in Tab. 1, we compare our proposed SDT-Net with state-of-the-art MOT methods on the SDTD test set. Our method outperforms previous approaches in tracking space debris, particularly under dense debris conditions. Especially, SDT-Net significantly outperforms CenterTrack in tracking performance. On the debris test set, it improves MOTA and HOTA by 10.6% and 6.8%, respectively, while on the dense debris test set, the improvements are 5.5% and 7.5%. These results demonstrate that SDT-Net achieves superior object association (higher AssA) and detection robustness (higher DetA). SDT-Net consistently outperforms CenterTrack across all key metrics, highlighting its effectiveness in complex space debris tracking scenarios.

Table 2: Ablation study of SDT-Net.

| | LSDM | RoI-FE | Offset Module | Debris Test ($n \leq 2$) | | | | Dense Debris Test ($1 \leq n \leq 5$) | | | |
|---|---|---|---|---|---|---|---|---|---|---|---|
| | | | | IDF1↑ | MOTA↑ | HOTA↑ | DetA↑ | IDF1↑ | MOTA↑ | HOTA↑ | DetA↑ |
| (a) | | | ✓ | 86.3 | 77.1 | 81.0 | 85.6 | 72.5 | 64.8 | 66.1 | 76.7 |
| (b) | ✓ | | ✓ | 90.4 | 85.5 | 84.8 | 88.3 | 78.5 | 64.9 | 65.3 | 79.1 |
| (c) | ✓ | ✓ | | 88.2 | 86.0 | 85.2 | 88.9 | 73.5 | 62.9 | 63.0 | 78.6 |
| (d) | ✓ | ✓ | ✓ | **91.8** | **87.7** | **87.8** | **90.9** | **80.6** | **70.3** | **73.6** | **81.6** |

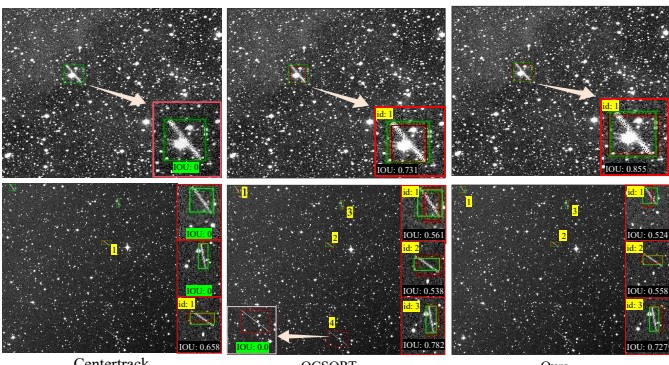

Figure 4: Visualization example of the tracking results of the SDTD model on the test set. The green bounding box represents the GT, and the red bounding box represents the model prediction result.

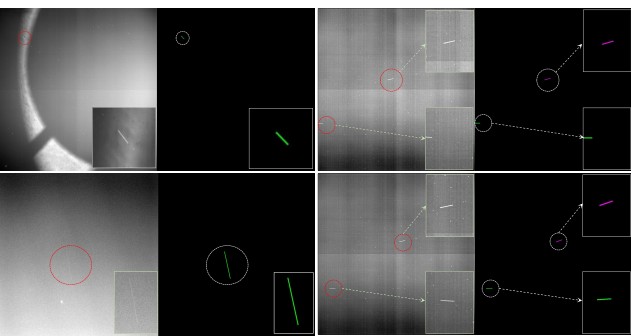

Figure 5: Visualization of sample tracking results from real-world observations. Right: model predicts tracking results, represented by predicted endpoints and connecting segments forming the trajectory (superimposed on a blank background). Different colors correspond to different IDs.

## 5.4 ABLATION STUDY

To analyze the contributions of different components in SDT-Net, we conduct an ablation study, as shown in Tab 2. We evaluate three configurations: LSDM (Line Source Detection Module), RoI-FE (Region of Interest Feature Enhancement) and Offset Module.

**LSDM**:Compared to CenterTrack (row a), introducing line source detection in SDT-Net (row b) brings notable gains across all metrics. MOTA rises from 77.1% to 85.5% on the Debris Test set and from 64.8% to 64.9% on the Dense Debris Test set. This indicates that modeling debris as segments rather than center-based boxes enhances spatial localization, improving detection (higher DetA) and tracking performance.

**RoI-FE**: Further incorporating RoI-FE (compare rows b and d) enhances the representation of debris features. This results in additional performance gains, with MOTA improving from 85.5% to 87.7% on the Debris Test set and from 64.9% to 70.3% on the Dense Debris Test set. Similarly, HOTA sees a significant increase, confirming that segmentation helps improve tracking robustness.

**Offset Module**: As shown in rows (c) and (d), introducing the debris offset module alone can significantly improve all indicators. Specifically, MOTA is improved from 86.0% to 87.7% on the debris test set, and from 62.9% to 70.3% on the dense debris test set. These improvements show that the offset module plays a key role in improving endpoint accuracy and overall tracking robustness.

## 5.5 QUALITATIVE RESULTS

We compare SDT-Net with OCSORT and CenterTrack. As shown in Fig. 4, under challenging conditions such as occlusion, dense star fields, and multiple debris, CenterTrack fails to maintain

tracking when debris is occluded by light sources, while SDT-Net successfully detects and tracks the debris. In dense debris scenarios, OCSORT produces false positives, whereas SDT-Net demonstrates superior spatial detection and tracking performance, accurately associating and localizing debris even in complex scenes.

To further validate the effectiveness of SDT-Net, we evaluate the model using real-world debris data. Due to the scarcity of real-world debris data, we ultimately collected 36 video sequences containing debris (a total of 2,228 frames) and invited astronomy experts to annotate them. As shown in Table 3, our method (SDT-Net) achieves a MOTA score exceeding 73%, demonstrating strong performance on real-world data. Fig. 5 further visualizes the results.

Table 3: Evaluation of SDT-Net on the Antarctic Station real-world dataset.

| Metric | Deepsort | Motdt | CTracker | Sort+YoloX | CenterTrack | ByteTrack | OCSORT | **SDT-Net** |
|--------|----------|-------|----------|------------|-------------|-----------|--------|-------------|
| MOTA↑ | 48.7 | 47.4 | 51.9 | 57.4 | 64.7 | 66.8 | 69.3 | **73.2** |
| HOTA↑ | 48.9 | 45.2 | 45.8 | 57.4 | 58.3 | 62.4 | 66.8 | **68.1** |
| DetA↑ | 51.8 | 47.6 | 54.5 | 58.9 | 68.7 | 70.1 | 72.2 | **75.8** |

## 6 CONCLUSION

In this paper, we introduce SDTD, the first benchmark dataset for tracking space debris. SDTD consists of 18,040 videos from ZTF with 62,562 frames and 250,000 synthetic debris. Based on this, we propose SDT-Net, which achieves state-of-the-art performance. In addition, we also evaluate the generalization ability of the model through real-world data. The results show that SDT-Net exhibits good tracking performance, proving that the model is transferable to distributed external scenes, ensuring strong tracking capabilities in challenging environments.

ETHICS STATEMENT

The authors acknowledge their responsibility to adhere to the ICLR Code of Ethics.

REPRODUCIBILITY STATEMENT

To ensure the reproducibility of our results, we provide comprehensive details of our methodology, experiments, and implementation.

- **Code and Data Availability:** We commit to making all source code, custom datasets, and data preprocessing scripts publicly available upon acceptance of this paper. The materials will be hosted in a public repository under a permissive open-source license to ensure full reproducibility and to facilitate future research.
- **Implementation Details:** A full description of our model architectures, algorithms, and experimental setup is provided in Appendix.

We believe this provides sufficient information for the research community to reproduce and build upon our findings.

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

## A   THE USE OF LARGE LANGUAGE MODELS (LLMs)

We acknowledge the use of a large language model (LLM) to improving grammar and wording of our paper. The authors are fully responsible for the content of this work.

## B   APPENDIX

### B.1   COMPLEX SKY LIGHT BACKGROUND

As shown in Fig. 6, the ZTF data contains complex skylight background. Below, we provide a detailed description of some of the background features illustrated in the figure:

**(Row 1, column 2)**: In this part of the image, we can see a dense star field. Most of the light points in the image are likely to be the light of stars or distant galaxies. Due to the large number of stars and galaxies, their light points are densely distributed on the image, which brings additional challenges to the identification and measurement of target celestial bodies. buraba **(Row 1, column 3)**: This image shows significant background brightness unevenness. There is a clear bright area in the upper left corner, which may be caused by the halo effect caused by a bright light source (such as a star or planet) at the edge of the field of view when shooting.

**(Row 2, column 4)**: The image shows the background feature of longitudinal stripes. Several vertical white stripes can be seen running through the entire image, which may be caused by defects in the camera sensor or electronic noise. Such longitudinal stripes will interfere with the clarity of the image and the identification of target celestial bodies.

**(Row 3, column 4)**: In this part of the image, the bright spots are likely to be the light of stars or distant galaxies, which is one of the common features in ZTF data. These bright spots represent light sources in the universe, from nearby stars to extremely distant galaxies, each with a different brightness and color that reflects their essential properties and distance from Earth.

### B.2   DATASET COMPARISON

As shown in Tab 4, existing space debris datasets are mostly small-scale images or videos with limited object categories and trajectory information. In contrast, the SDTD dataset contains both large-scale synthetic and real data, introduces multi-object tracking annotations, and provides a richer set of object types, providing more comprehensive support for subsequent detection and tracking research.

Table 4: Summary of Space Debris Target Datasets.

| Dataset | Type | Size | #Video. | #Det. | #Track. | Target Variations | Debris Number |
|---|---|---|---|---|---|---|---|
| Lin et al. (2021) | Real | 83 images | 7 videos | ✓ | × | One | One |
| SDebrisNet (Tao et al. 2023) | Real | 1551 images | × | ✓ | × | Three | One |
| Jiang et al. (2022b) | Real | 200 images | × | ✓ | × | One | One |
| AstroStripeSet Zhu et al. (2024) | Real | 1500 images | × | ✓ | × | Four | One |
| **SDTD** | Synthetic & Real | 62,562 & 2,228 images | 18,040 & 36 videos | ✓ | ✓ | Five | One to Five |

### B.3   DATASET STATISTICS

We first collect 16,040 ZTF astronomical observation data samples, and simulate each of them to synthesize a video, thereby constructing a training-validation set consisting of 16,040 videos. To evaluate model performance in more challenging scenarios, we randomly select 1,000 samples from this dataset and simulate them further to construct two separate test sets: the debris test set and the dense debris test set.

After all simulations, we obtain a total of 18,040 videos comprising 62,562 frames. As shown in Fig. 7(a), the two test sets are designed to comprehensively assess the model's robustness in different environments, particularly under conditions with varying levels of debris density.

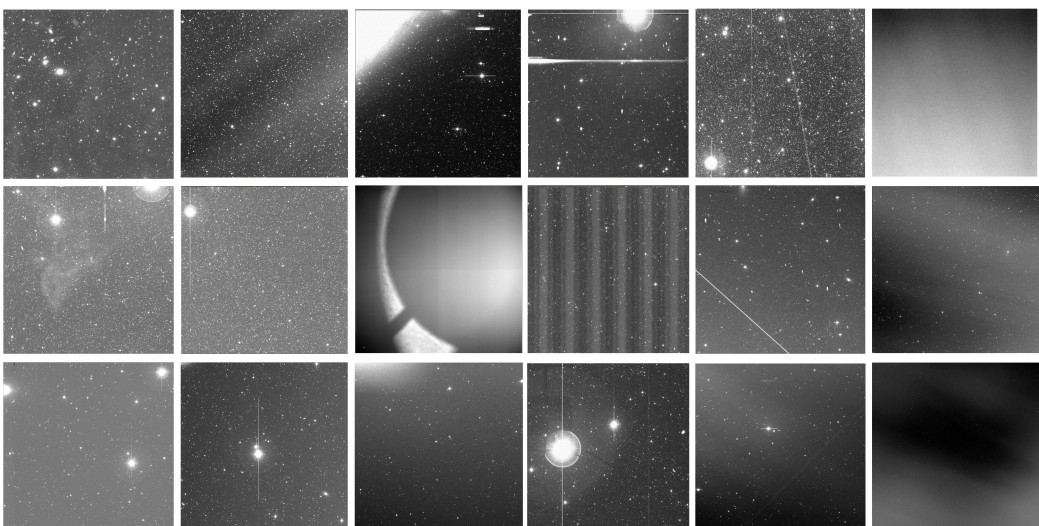

Figure 6: Data example of complex skylight background.

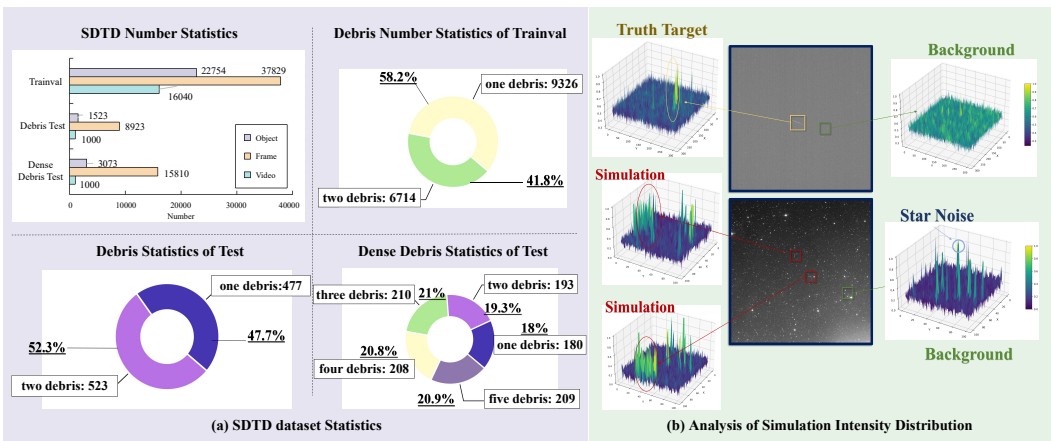

Figure 7: Overview of SDTD dataset statistics and analysis of debris simulation intensity distribution. (a) The left shows the number statistics for training, test, and dense debris categories in the SDTD dataset. The pie charts represent the distribution of debris across different objects in both the training and test sets. (b) The right displays the analysis of simulation intensity distribution, comparing the target, simulation, background, and star noise. It also shows the intensity distribution for debris and background noise.

## B.4   DEBRIS TYPE

To enable a comprehensive evaluation of model performance under varying levels of complexity, we categorize the space debris in our dataset into two types: debris and dense debris. This type allows for a systematic assessment of tracking robustness in both sparse and cluttered observational scenarios.

The debris type refers to space objects that appear individually or sparsely within a frame. These objects are well-separated in space and time, exhibiting minimal overlap, which makes them relatively easier to detect and track. In contrast, the dense debris type characterizes scenes with a high concentration of targets—typically more than two debris objects appearing within a single frame or across adjacent frames. Such scenes often involve significant spatial overlap and motion correlation between objects, posing greater challenges for identity preservation and temporal association.

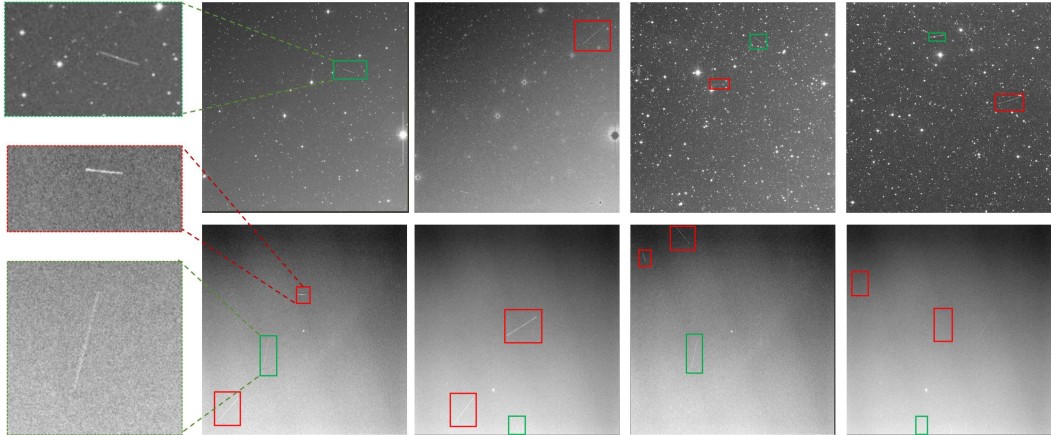

Figure 8: Comparison of debris simulation and real-world debris. The green bounding box represents real-world debris, and the red bounding box represents simulated debris.

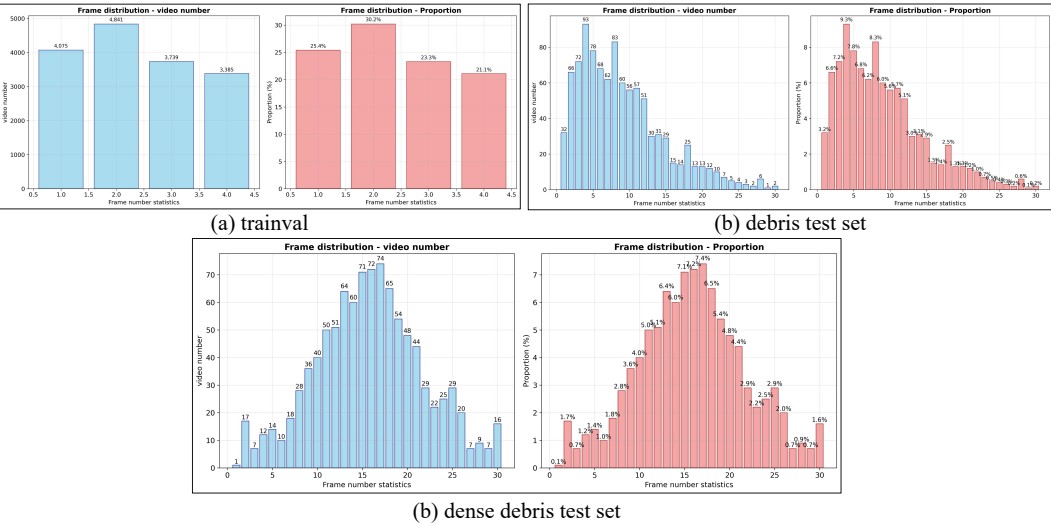

(a) trainval

(b) debris test set

(b) dense debris test set

Figure 9: Frame distribution statistics for the train-val, debris test, and dense debris test sets.

### B.5 SIMULATION VISUALIZATION

Line source debris in the image typically exhibits high-intensity values. To better compare the effects of simulated line source debris, we visualize the intensity distribution analysis, where different regions of the image are displayed in 3d on the left.

The yellow box highlights the intensity distribution of a real-world debris, revealing distinct concentrated peaks that represent the characteristic signal intensity of an actual celestial body. These peaks indicate localized regions where the target signal is densely distributed.

In contrast, the red box corresponds to a simulated debris. Its intensity distribution follows a similar trend, demonstrating that the simulated data closely approximates the structural characteristics of real observations. Additionally, the green box illustrates the background noise distribution. In the simulated image, this background (e.g., star field noise) exhibits multiple peaks, indicating a more dispersed signal distribution and the presence of occasional strong interference.

By comparing these distributions, we validate the effectiveness of the simulation process, confirming that the simulated data retains the essential structural features of real targets, even under noisy conditions.

To provide a more intuitive understanding of the results, we also visualize the simulated debris directly, as shown in Fig 8.

#### B.5.1 VIDEO ANALYSIS

To understand the distribution of video lengths in our dataset, we conduct a statistical analysis of the number of frames per video across the training-validation and test sets, as illustrated in Fig. 9. The train-val set primarily consists of short video clips, with most videos containing 1 to 4 frames. In contrast, the debris test set shows a much wider distribution, with video lengths ranging from 1 to 30 frames, reflecting a higher degree of variation. The dense debris test set further differs in that its video lengths are more concentrated around 18 frames, indicating a consistently longer temporal span in complex scenes.

This design is on purpose and has clear benefits. The shorter video clips in the train-val set offer a simple and focused training setup, which makes it easier to fine-tune the model. With fewer frames to process, the model can quickly learn the essential tracking behaviors and adapt to various scenarios. This also facilitates faster experimentation and parameter tuning. On the other hand, the longer and more variable video lengths in the test sets, especially in the dense debris scenario, are designed to better reflect real-world observational conditions, where targets often persist across multiple frames and exhibit complex motion patterns. This contrast enables a more realistic and robust evaluation of model performance across a range of temporal complexities.

### B.6 EXPERIMENTAL SETTING/DETAILS

We ensure reproducibility by providing the experimental environment and computational resources. Tab. 5 shows the environment configuration.

The entire model is trained for 60 epochs, and the initial learning rate is set to 3e-3, which is decayed by 0.1 at the 20 epoch. Additionally, We adopt a downsampling strategy to generate paired images for training. Specifically, we first crop image patches of size $1524 \times 1524$ from the original high-resolution images as ground truth. We employ the resolution of the input images is set to $1524 \times 1524$ and the batch size is set to 2. Besides, the value of $\lambda_{\text{seg}}$ and $\lambda_{\text{emb}}$ are set to 1.0, $\lambda_{\text{hm}}$ is set to 10, and $\lambda_{\text{off}}$ is set to 0.1. Notably, the tracking methods compared in **Tab. 2** of **Sec. Experiments** in the main manuscript are all retrained using SDTD according to the official settings.

#### B.6.1 SIMULATION PARAMETERS

The simulation parameters are listed below, with each variation and its corresponding range:

Table 5: Experimental Environment Setup.

| Component | Version |
|---|---|
| OS | Ubuntu 20.04.5 LTS |
| Python | 3.10.15 |
| PyTorch | 2.0.0 |
| CPU | Intel Xeon Platinum 8558 |
| CUDA | 11.8 |

Table 6: Table showing variations and their respective parameters.

| Variation | Parameter (Range) |
|---|---|
| Number | 1, 5 |
| Initial point x | 0, 1524 |
| Initial point y | 0, 1524 |
| Length | 40, 480 |
| Width | 12, 32 |
| Angle | 0, 360 |
| Fracture | 0, 2 |
| Displacement | 50, 700 |
| Noise | 20, 100 |
| PSF | mean: width/2, $\delta_{psf}$: width/8, $S$: (20, 120) |

## B.7 EXPERIMENTS AND VISUALIZATIONS

### B.7.1 REAL-TIME ANALYSIS

To evaluate the efficiency of SDT-Net, we measured the inference time for a single image. The results are shown in Tab. 7. The model achieves an inference time of 0.37 seconds per image, satisfying the performance requirements for real-time applications.

Table 7: Evaluate the inference time of each method on a single image.

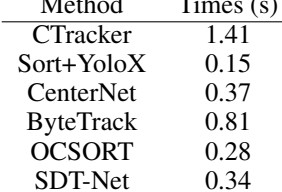

| Method | Times (s) |
|---|---|
| CTracker | 1.41 |
| Sort+YoloX | 0.15 |
| CenterNet | 0.37 |
| ByteTrack | 0.81 |
| OCSORT | 0.28 |
| SDT-Net | 0.34 |

### B.7.2 STATISTICS OF FALSE POSITIVES AND FALSE NEGATIVES

To provide a clearer understanding of SDT-Net's error behavior, we additionally report the number of false positives and false negatives on both test sets. These statistics complement the tracking metrics presented in the main paper and offer a more detailed information of detection errors in different debris densities.

As shown in Table 8, the dense debris test set exhibits a higher number of false positives and false negatives due to severe object overlap, cluttered star fields, and the presence of multiple visually similar debris streaks.

### B.7.3 ADDITIONAL COMPARISONS WITH TRACKING MODELS

To provide a more comprehensive comparison across modern tracking paradigms, we additionally evaluate transformer-based trackers, joint detection–embedding models, and several recent tracking methods introduced after 2024. The results are summarized in Table 9.

Table 8: False positives (FP) and false negatives (FN) on the debris and dense debris test sets.

| Test Set | FP | FN |
|---|---|---|
| Debris test | 333 | 596 |
| Dense debris test | 1698 | 1928 |

Table 9: Comparison with transformer-based trackers, joint detection–embedding models, and recent tracking methods on the Space Debris and Dense Debris test sets.

| | Space Debris | | | | Dense Debris | | | |
|---|---|---|---|---|---|---|---|---|
| Method | MOTA | HOTA | DetA | AssA | MOTA | HOTA | DetA | AssA |
| MOTR Zeng et al. (2022) | 70.4 | 68.8 | 72.1 | 65.5 | 56.8 | 54.9 | 56.5 | 53.3 |
| TrackFormer Meinhardt et al. (2022) | 77.0 | 71.2 | 76.5 | 68.6 | 60.2 | 71.3 | 67.5 | 54.6 |
| FairMOT Zhang et al. (2021) | 72.5 | 70.2 | 74.8 | 66.1 | 52.4 | 55.0 | 59.2 | 53.5 |
| PD-SORT Wang et al. (2025) | 78.9 | 80.1 | 82.4 | 78.0 | 64.8 | 67.3 | 69.1 | 64.2 |
| Hybrid-SORT Yang et al. (2024) | 84.5 | 85.6 | 85.1 | 86.0 | 68.7 | 72.1 | 74.5 | 69.7 |
| SDT-Net | 87.7 | 87.8 | 90.9 | 84.8 | 70.3 | 73.6 | 81.6 | 65.6 |

### B.7.4 Loss and Backbone ablation experiment

To further verify the sensitivity of the loss weights, we conducted additional experiments by changing the hyperparameters ($\lambda_{hm}$, $\lambda_{seg}$, $\lambda_{emb}$, and $\lambda_{off}$). As shown in the tables below, our default setting achieves the best performance:

Tab. 11 compares different backbone architectures. DLA34 achieves the best results, outperforming ResNet50 and Hourglass-104 in both detection and association metrics, demonstrating its superior suitability for debris tracking. Moreover, the alternative backbones do not benefit from the DLA-Up aggregation module, which further contributes to the performance gap.

### B.7.5 Cross-Domain Evaluation

To further evaluate transferability under different observational setups, we synthesized three additional datasets using backgrounds from HST, SDSS, and ZTF. We directly applied the pre-trained SDT-Net for inference. The results are summarized in Tab. 12. These results not only demonstrate the effectiveness of SDT-Net, but its strong cross-domain performance further confirms that the real-data evaluation is reliable.

Table 10: Sensitivity analysis of loss-weight hyperparameters. The default setting for each term is indicated as _.

| Hyperparameter | Value | MOTA ↑ | HOTA ↑ | DetA ↑ | AssA ↑ |
|---|---|---|---|---|---|
| $\lambda_{hm}$ | 0.1 | 77.7 | 77.8 | 80.1 | 75.5 |
| | 1.0 | 85.4 | 84.3 | 85.2 | 83.4 |
| | 10.0 | 87.7 | 87.8 | 90.9 | 84.8 |
| $\lambda_{seg}$ | 0.1 | 84.4 | 83.1 | 86.9 | 79.3 |
| | 0.5 | 86.8 | 85.7 | 88.2 | 83.2 |
| | 1.0 | 87.7 | 87.8 | 90.9 | 84.8 |
| $\lambda_{emb}$ | 0.1 | 85.1 | 85.5 | 88.7 | 82.3 |
| | 1.0 | 87.7 | 87.8 | 90.9 | 84.8 |
| $\lambda_{off}$ | 1.0 | 86.9 | 87.3 | 90.1 | 84.4 |
| | 0.1 | 87.7 | 87.8 | 90.9 | 84.8 |

Table 11: Comparison of different backbone architectures for SDT-Net. DLA34 achieves the best overall performance.

| Backbone | MOTA ↑ | HOTA ↑ | DetA ↑ | AssA ↑ |
|---|---|---|---|---|
| ResNet50 | 82.6 | 80.1 | 86.6 | 74.6 |
| Hourglass-104 | 86.4 | 85.6 | 88.7 | 83.5 |
| DLA34 (ours) | 87.7 | 87.8 | 90.9 | 84.8 |

Table 12: Cross-domain evaluation of SDT-Net on datasets using backgrounds from HST, SDSS, and ZTF.

| Data Source | MOTA ↑ | HOTA ↑ | DetA ↑ | AssA ↑ |
|---|---|---|---|---|
| HST (F814W) | 66.7 | 64.5 | 67.2 | 61.9 |
| SDSS (r-band) | 67.4 | 69.4 | 71.9 | 66.9 |
| ZTF (g-band) | 69.7 | 72.1 | 74.9 | 69.5 |
| SDTD (dataset) | 70.3 | 73.6 | 81.6 | 65.6 |

### B.7.6 ANALYSIS ON CLIP LENGTH AND LONG-SEQUENCE TRAINING

To examine whether training on very short video clips restricts the model's ability to learn temporal cues, we conduct an additional experiment using longer training sequences. Most training and validation clips contain only 1 to 4 frames, while the test sets include sequences that can reach 30 frames, and the dense test set typically contains sequences with an average length close to 18 frames.

We additionally generated a set of long-sequence debris videos for training, consisting of 2,500 videos with 30,500 frames (average length 12 frames). To keep the overall training scale comparable to the original setting, we mixed these with 3,500 videos randomly sampled from the original training set. The combined dataset contains 6,000 videos with 38,760 frames, which is similar in size to the original training set ( 37,829 frames). We re-trained SDT-Net using this combined dataset and summarize the results in Table 13.

We observed a significant improvement in performance for dense and long segments. In our future work, we will expand the dataset in this way to further improve model performance.

### B.7.7 RESULTS VISUALIZATION

We provide a detailed visualization of the results. As shown in Fig. 10, we present the tracking result of SDT-Net on real-world data. Both Exp1 and Exp2 demonstrate strong tracking performance on debris, indicating that our model exhibits good generalization capabilities for real-world scenarios.

Furthermore, as shown in Fig. 11 and Fig. 12, we visualize the results on the SDTD test set, providing further evidence of the model's robustness across different test conditions. Fig. 13 visualizes the results on the SDTD dense test set, where the performance highlights the model's ability to effectively handle complex, densely cluttered scenarios.

Table 13: Evaluation of SDT-Net trained with short sequences and with long-sequence augmented training data.

| Training Setting | MOTA ↑ | HOTA ↑ | DetA ↑ | AssA ↑ |
|---|---|---|---|---|
| Original training set | 70.3 | 73.6 | 81.6 | 65.6 |
| Long-sequence augmented set | 73.9 | 74.9 | 82.9 | 67.5 |

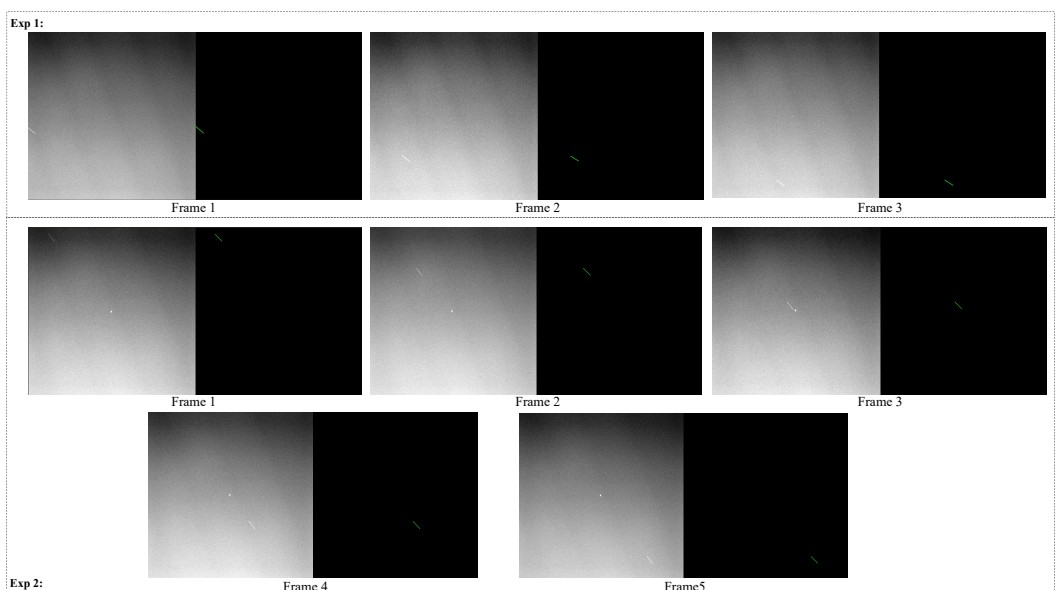

Figure 10: SDT-Net tracking results in the real-world data.

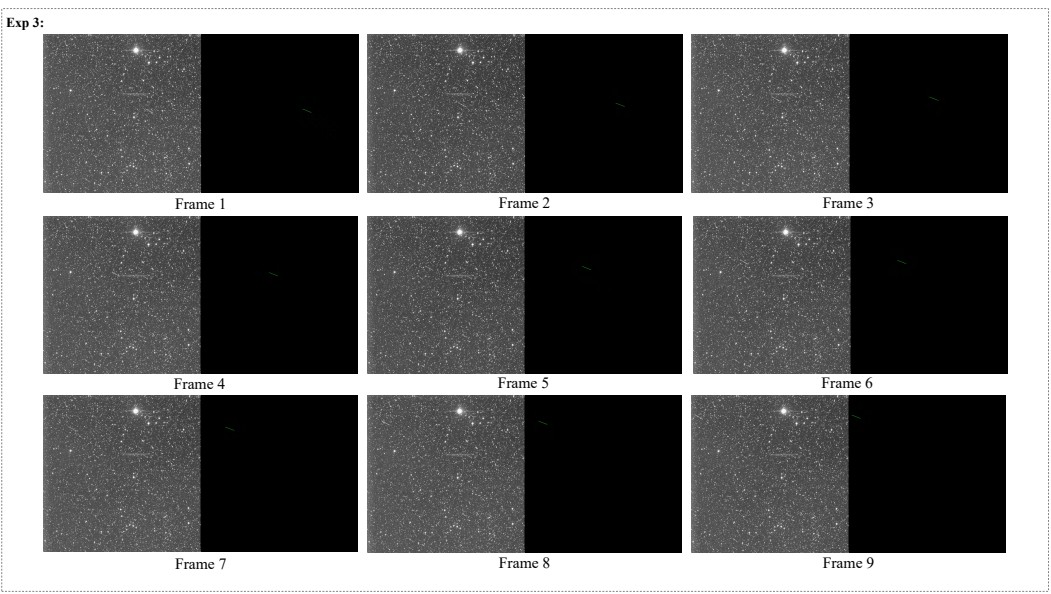

Figure 11: SDT-Net test tracking result example in SDTD.

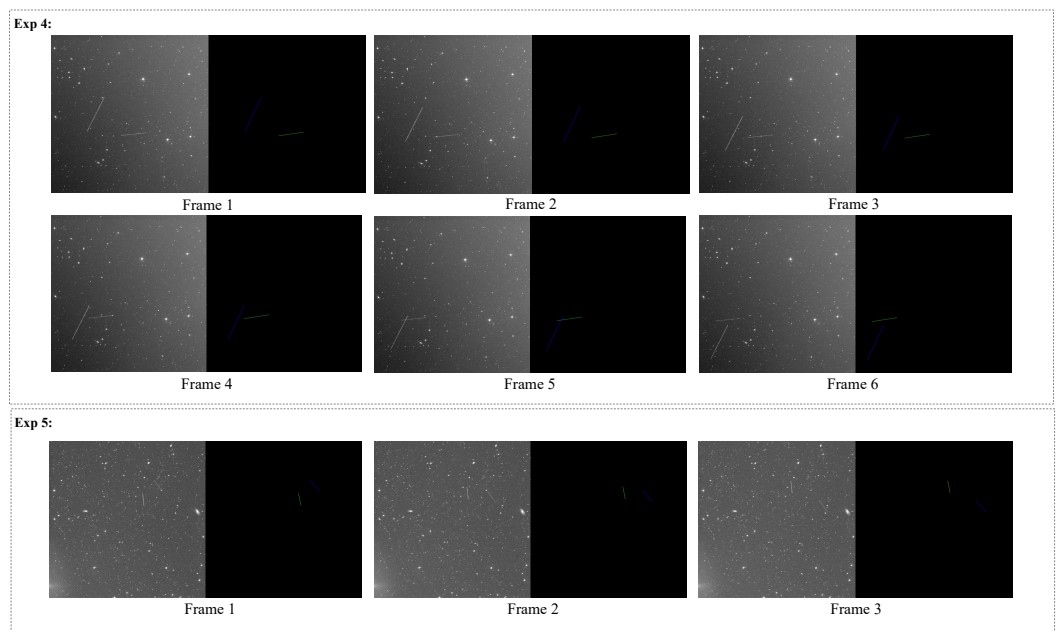

Figure 12: Example of tracking results of SDT-Net on the test set.

### B.7.8 DISCUSSION OF INTEGRATION WITH ASTRONOMICAL PROCESSING PIPELINES

We discuss how the outputs of SDT-Net can be directly integrated into standard astronomical data-processing pipelines for debris-related tasks. It is straightforward to integrate our system into existing astronomical processing pipelines. In the current debris processing pipeline, a debris-detection module should take a FITS (an image format in astronomy) image as input and output the sky coordinates of each detected object. Our model follows the same interface: it is trained on raw FITS-domain images, so it has the ability to directly operate FITS images. And its predicted pixel-level locations can be converted into standard sky coordinates (Right Ascension and Declination) using the WCS (World Coordinate System) metadata stored in every FITS file. This means the outputs of SDT-Net are immediately usable by downstream astronomy modules—such as orbit estimation, trajectory prediction, and catalogue updating—without any additional transformation or engineering effort.

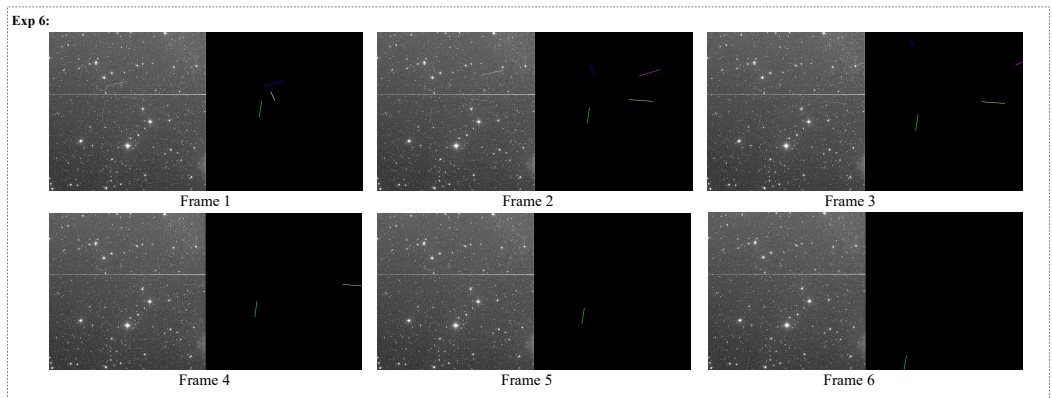

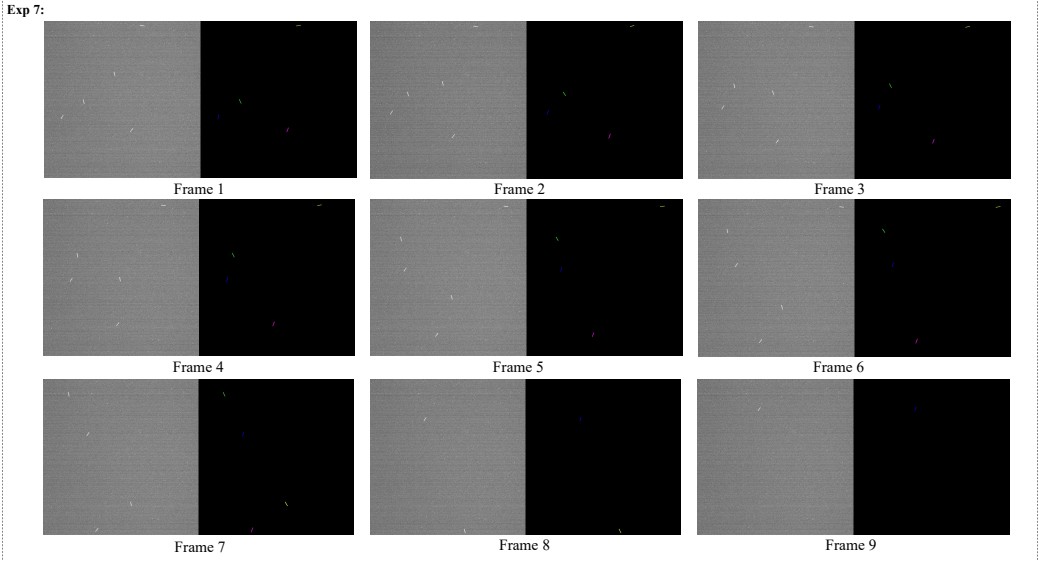

Figure 13: Example of tracking results of SDT-Net on the dense test set.

