# OpenReview forum: "High Performance Space Debris Tracking in Complex Skylight Backgrounds with a Large-Scale Dataset"
_ICLR.cc/2026/Conference — Submitted to ICLR 2026_

### Official Review · Reviewer_k1G9 · 2025-10-26

**Soundness:** 3
**Presentation:** 3
**Contribution:** 3
**Rating:** 6
**Confidence:** 3

**Summary:**

This paper introduces SDT-Net, a deep-learning-based tracker for space debris, and SDTD, a large-scale synthetic dataset for training/evaluation. The authors simulate 18k videos (62k frames) of linear-shaped debris on real ZTF backgrounds, then benchmark SDT-Net against MOT methods. An ablation study confirms the utility of the proposed RoI-FE module, line-source detection head, and debris-offset association mechanism.

**Strengths:**

1. Automated debris tracking is critical for space-safety, and data-hungry DL approaches have been stymied by the absence of large annotated corpora.
2. SDTD is two orders of magnitude larger than prior debris data and includes both synthetic and real labels; the simulation pipeline (PSF convolution, realistic motion, ZScale preprocessing) is well motivated and reproducible.

**Weaknesses:**

1. Recent transformer trackers (e.g., MOTR, TrackFormer) and joint detection-embedding models (e.g., FairMOT) are ignored. The authors should at least include one modern transformer baseline or justify its exclusion.
2. The performance improvement compared with OCSORT is minor. More importantly, the proposed method is trained on the debris tracking dataset, while the OCSORT is trained on the general video. It is hard to identify whether the performance improvement comes from the method or the training data.

**Questions:**

I am not an expert on debris tracking. I encourage the authors to clarify the Weaknesses. The proposed benchmark seems valuable.

---

> ### Author Response · Authors · 2025-11-21
>
> Thanks for your valuable feedback and efforts.
>
> **1. Recent transformer trackers (e.g., MOTR, TrackFormer) and joint detection-embedding models (e.g., FairMOT) are ignored. The authors should at least include one modern transformer baseline or justify its exclusion.**
>
> Thank you for your suggestion. In response, we have included additional comparisons with transformer-based trackers such as MOTR and TrackFormer, as well as joint detection-embedding model such as FairMOT. We further add several recent tracking methods introduced after 2024, including PDSORT and Hybrid-SORT.
>
> The results are as follows:
> |  Space Debris | MOTA | HOTA  | DetA | AssA |
> | -------------- | -------- | -------- | -------- |-------- |
> | MOTR     |  70.4     |  68.8    | 72.1  |65.5 |
> | TrackFormer | 77.0  |  71.2   | 76.5  | 68.6   |
> | FairMOT |  72.5  |  70.2    |  74.8   |  66.1   |
> | PDSORT |   78.9  |  80.1   |   82.4   | 78.0  |
> | Hybrid-SORT |   84.5  |  85.6   |  85.1    |  86.0   |
> | SDT-Net |  87.7     |  87.8     |  90.9  |84.8 |
>
>
> |  Dense Debris | MOTA | HOTA  | DetA | AssA |
> | -------------- | -------- | -------- | -------- |-------- |
> | MOTR     |  56.8     |  54.9     |  56.5  |53.3 |
> | TrackFormer |  60.2 |  71.3   |  67.5 |  54.6  |
> | FairMOT |  52.4   |   55.0   |   59.2  | 53.5   |
> | PD-SORT |  64.8   | 67.3    |   69.1   | 64.2  |
> | Hybrid-SORT |  68.7   |  72.1   |  74.5    |  69.7   |
> | SDT-Net |  70.3     |  73.6     |  81.6  |65.6 |
>
>
> Compared with other methods, our model still achieves the best result, showing strong potential in the space-debris tracking scenario.
>
>
> **2. The performance improvement compared with OCSORT is minor. More importantly, the proposed method is trained on the debris tracking dataset, while the OCSORT is trained on the general video. It is hard to identify whether the performance improvement comes from the method or the training data.**
>
> We clarify that to ensure a fair comparison, all methods in Table 1 (including OCSORT) were fully retrained and evaluated on the SDTD dataset under the same official settings. This is explicitly stated in Appendix B.6: “Experiments in the main manuscript are all retrained using SDTD according to the official settings.”

---

### Official Review · Reviewer_AqoJ · 2025-10-30

**Soundness:** 2
**Presentation:** 3
**Contribution:** 2
**Rating:** 4
**Confidence:** 4

**Summary:**

This manuscript proposes SDT-Net, a deep learning-based network for space debris tracking, designed to achieve high-precision, real-time debris monitoring. The authors concurrently constructed a large-scale simulated dataset, SDTD, for model training and evaluation. Experimental results demonstrate that SDT-Net achieves exceptional performance in both synthetic and real-world scenarios (utilizing data from the Antarctic station), attaining 73.2% MOTA on real-world data, which validates its strong generalization capability and practical utility.

**Strengths:**

Motivation and Meaning: motivation is feasible
This manuscript focuses on detecting and predicting the motion trajectories of space debris to mitigate collision risks and advance the development of the aerospace industry.

Dataset：
This manuscript constructs a relatively large-scale dataset for space debris tracking, addressing a critical gap in existing research resources.

Writing Quality：
The manuscript is clearly articulated and highly readable. The methodological approach, experimental validation, and visualizations are comprehensively presented and well-supported.

**Weaknesses:**

Innovation: the contribution not enough for a ICLR paper
The design of RoI-FE demonstrates the distinction between SDT tasks and classical detection and tracking methods, suggesting that solutions tailored to the unique challenges of SDT may be necessary.


Methodology Section:
The explanation of the final loss function is insufficient, and there is a lack of corresponding ablative experimental analysis.

Experimental Evaluation:
The comparative experiments and ablative studies are inadequate.

**Questions:**

1. The comparative experiments lack comparisons with state-of-the-art methods published after 2024, and also lack comparisons with backbone methods.
2. What are the fundamental distinctions between the SDT task and classical detection and tracking methods, and what are its unique challenges?
3. RoI-FE is a straightforward feature fusion module composed of multiple stacked convolutional layers. What is the computational cost associated with this operation?

---

> ### Author Response · Authors · 2025-11-21
> **Reply to Weaknesses**
>
> Thanks for your valuable feedback and efforts.
>
> **Weaknesses:**
>
> **1. Innovation: the contribution not enough for a ICLR paper The design of RoI-FE demonstrates the distinction between SDT tasks and classical detection and tracking methods, suggesting that solutions tailored to the unique challenges of SDT may be necessary.**
>
> The contribution of this submission is not limited to RoI-FE but also includes the dataset itself. Specifically: (1) we are the first to define and establish the debris tracking task, whereas prior work has focused almost exclusively on detection. We have developed a complete debris simulation toolkit, constructed a large-scale synthetic dataset (SDTD), and annotated a set of real data. All of these will be released upon acceptance, and (2) based on these, we provide a simple yet effective model to validate the reliability of the dataset and to serve as a foundation for future community driven method development.
>
> Overall, our contribution lies in the combination of the dataset and the method as a whole, rather than being centered on the RoI-FE module alone.
>
> **2.Methodology Section: The explanation of the final loss function is insufficient, and there is a lack of corresponding ablative experimental analysis.**
>
> Thank you for the reviewer's comment.
>
> Our overall loss function (Equation16) is defined as::
>
> $L=\lambda_{seg}L_{seg}+\lambda_{hm}L_{hm}+\lambda_{emb}(L_{same}+L_{diff})+\lambda_{off}L_{off}$.
>
> This objective is a multi-task learning formulation, where:
>
> $L_{seg}$：Optimizes the RoI-EF module for pixel-level debris segmentation.
>
> $L_{hm}$: Supervisions the line source Heatmap generation for object endpoints localization.
>
> $L_{same},L_{diff}$: Constrains the LSDM module to learn embeddings for association.
>
> $L_{off}$: Refines the error in the Offset Module.
>
> Regarding the contribution of each component, the module-level ablation study is already provided in Table 2 of the main paper (demonstrating the necessity of LSDM, RoI-EF, and Offset modules).
>
> To further address the concern regarding the sensitivity of loss weights, we conducted additional experiments by varying the hyperparameters ($\lambda_{seg}$,$\lambda_{hm}$,$\lambda_{emb}$,$\lambda_{off}$). As shown in the tables below, our default setting (labeled "ours") achieves the best performance:
>
> | $\lambda_{hm}$ | MOTA | HOTA | DetA  | AssA |
> | -------------- | ---- | ---- | ---- | ---- |
> | 0.1            | 77.7 | 77.8 | 80.1 | 75.5 |
> | 1.0            | 85.4 | 84.3 | 85.2 | 83.4 |
> | 10.0 (ours)    | 87.7 | 87.8 | 90.9 | 84.8 |
>
> | $\lambda_{seg}$ | MOTA | HOTA | DetA  | AssA |
> | --------------- | ---- | ---- | ---- | ---- |
> | 0.1             | 84.4 | 83.1 | 86.9 | 79.3 |
> | 0.5             | 86.8 | 85.7 | 88.2 | 83.2 |
> | 1.0(ours)       | 87.7 | 87.8 | 90.9 | 84.8 |
>
>
> | $\lambda_{emb}$ | MOTA | HOTA | DetA  | AssA |
> | --------------- | ---- | ---- | ---- | ---- |
> | 0.1             | 85.1 | 85.5 | 88.7 | 82.3 |
> | 1.0(ours)       | 87.7 | 87.8 | 90.9 | 84.8 |
>
>
> | $\lambda_{off}$ | MOTA | HOTA | DetA  | AssA |
> | --------------- | ---- | ---- | ---- | ---- |
> | 1.0             | 86.9 | 87.3 | 90.1 | 84.4 |
> | 0.1(ours)       | 87.7 | 87.8 | 90.9 | 84.8 |
>
> We will include these detailed definitions and sensitivity analyses in the Rebuttal Revision.
>
> **3.Experimental Evaluation: The comparative experiments and ablative studies are inadequate.**
>
> Our experiments cover as much as possible what we know, including: (1) comparative evaluations against multiple existing methods (e.g., CenterTrack, OCSORT) in Table 1 of the main paper, (2) ablation studies for each proposed module in Table 2, (3) loss-weight sensitivity analysis provided in our response to Comment 2, (4) additional comparisons with post-2024 methods included in our response to Question 1, and (5) ablations on different backbones, also provided in Question 1.
>
> If you have any another insights or suggestions, we would be happy to incorporate them and supplement our experiments in the final revision.

---

> > ### Author Response · Authors · 2025-11-21
> > **Reply to Questions**
> >
> > **Questions:**
> >
> > **1.The comparative experiments lack comparisons with state-of-the-art methods published after 2024, and also lack comparisons with backbone methods.**
> >
> > We have supplemented the comparative evaluation with state-of-the-art methods published after 2024, as well as ablations on different backbones. The additional results are summarized below：
> >
> > |  Space Debris | MOTA | HOTA  | DetA | AssA |
> > | -------------- | -------- | -------- | -------- |-------- |
> > | PDSORT(2024) |   78.9  |  80.1   |   82.4   | 78.0  |
> > | Hybrid-SORT(2024) |   84.5  |  85.6   |  85.1    |  86.0   |
> > | SDT-Net |  87.7     |  87.8     |  90.9  |84.8 |
> >
> > |  Dense Debris | MOTA | HOTA  | DetA | AssA |
> > | -------------- | -------- | -------- | -------- |-------- |
> > | PD-SORT(2024) |  64.8   | 67.3    |   69.1   | 64.2  |
> > | Hybrid-SORT(2024) |  68.7   |  72.1   |  74.5    |  69.7   |
> > | SDT-Net |  70.3     |  73.6     |  81.6  |65.6 |
> >
> > These results show that SDT-Net remains highly competitive against recent SOTA trackers, particularly in the space-debris setting.
> >
> > | Backbone      | MOTA | HOTA | Det  | AssA |
> > | ------------- | ---- | ---- | ---- | ---- |
> > | ResNet50      | 82.6 | 80.1 | 86.6 | 74.6 |
> > | Hourglass-104 | 86.4 | 85.6 | 88.7 | 83.5 |
> > | DLA34  (ours) | 87.7 | 87.8 | 90.9 | 84.8 |
> >
> > These experiments demonstrate that our chosen backbone provides the best accuracy.
> >
> > **2.What are the fundamental distinctions between the SDT task and classical detection and tracking methods, and what are its unique challenges?**
> >
> > From an astronomy perspective, the SDT task addresses a scientifically significant problem because long-exposure debris line source differ substantially from those considered in classical detection and tracking. As a result, existing vision-based methods are not directly applicable, and exploring this task therefore carries substantial scientific value.
> >
> > From a computer vision perspective, SDT also introduces several unique challenges:
> > (1) Extremely small source line: Debris often appears as very small objects or thin line sources with extreme aspect ratios and large shape variability, far beyond those in conventional tracking tasks.
> > (2) Weakly discriminative appearance: Low contrast and highly similar object appearance make them difficult to distinguish.
> > (3) Extreme signal-to-noise ratio: Such low signal-to-noise ratio conditions are uncommon in conventional natural-image settings.
> >
> > For astronomy, the dataset and method fill an important gap in debris observation. For computer vision, SDT provides a fundamentally different challenge and a new benchmark beyond traditional tracking tasks.
> >
> > **3.RoI-FE is a straightforward feature fusion module composed of multiple stacked convolutional layers. What is the computational cost associated with this operation?**
> >
> > We further conducted a computational cost analysis of the RoI-FE module. The results show that the  RoI-FE module contains approximately ~4M parameters and requires ~62 FLOPs(G), with an overall runtime of ~33 ms. RoI-FE accounts for only a small fraction of this cost, and its impact on the end-to-end inference speed is negligible. Therefore, we believe that RoI-FE achieves a reasonable performance cost trade-off for the current task.

---

### Official Review · Reviewer_y5Cr · 2025-11-01

**Soundness:** 3
**Presentation:** 3
**Contribution:** 2
**Rating:** 6
**Confidence:** 3

**Summary:**

The paper introduces SDT-Net, a CenterTrack-style tracker tailored to long-exposure, line-like space debris. The network adds a Region-of-Interest Feature Enhancement (RoI-FE) segmentation mask, endpoint heatmaps with a pairing embedding, and a per-endpoint offset head for frame association. The authors also build SDTD, a large synthetic-plus-real benchmark derived from ZTF backgrounds via an observation-based simulation with PSF blur and random truncation. SDTD comprises 18,040 videos / 62,562 frames / ~250k debris and is used to train/evaluate SDT-Net

**Strengths:**

1. Large, reproducible benchmark built from real survey backgrounds (ZTF) with PSF and truncation adds realism; the dataset scale and explicit dense-scene split are valuable to the community.

2. On SDTD, SDT-Net improves over CenterTrack/OCSORT/ByteTrack; on real Antarctic data it leads across MOTA/HOTA/DetA. Ablations isolate the gains from line-segment detection, RoI-FE, and the offset head.

3. Clear task formulation with architecture tweaks that match physics. Modeling debris as paired endpoints plus an offset field is well-motivated for line-sources; RoI-FE reduces skylight clutter before detection/association. The components (heatmap loss, CornerNet-style embedding push/pull, offset regression) are standard but effective.

**Weaknesses:**

1. Although the SDTD dataset is large and covers complex backgrounds, its generation process is mainly based on superimposing line sources from the background of ZTF astronomical images. The paper uses simple long-exposure line drawing with Gaussian blurring, without introducing high-fidelity physical constraints such as realistic trajectory dynamics modeling, PSF spatial variation, photosensitivity saturation, or noise field modeling. Therefore, from a technical perspective, the dataset's contribution leans more towards the scale of engineering collection and synthesis than proposing new physical fidelity or statistical generation mechanisms in simulation methods.

2. The overall architecture of SDT-Net largely follows the existing multi-object tracking paradigm: it centers on detection-regression-association, using heatmap regression endpoints, embedding matching, and offset prediction to achieve temporal correlation. The proposed "Region-of-Interest Feature Enhancement (RoIFE)" module and "offset module" are essentially lightweight integrations of existing feature enhancement and motion offset ideas, without introducing new mechanisms in the algorithm's principles or optimization objectives.

3. Train/val clips are mostly 1–4 frames, whereas test sets include sequences up to 30 frames (dense set concentrated around ~18). It’s unclear whether training on very short clips biases the tracker or underutilizes temporal cues.

**Questions:**

Please refer to the weaknesses

---

> ### Author Response · Authors · 2025-11-21
>
> Thanks for your valuable feedback and efforts.
>
> **1. Although the SDTD dataset is large and covers complex backgrounds, its generation process is mainly based on superimposing line sources from the background of ZTF astronomical images. The paper uses simple long-exposure line drawing with Gaussian blurring, without introducing high-fidelity physical constraints such as realistic trajectory dynamics modeling, PSF spatial variation, photosensitivity saturation, or noise field modeling. Therefore, from a technical perspective, the dataset's contribution leans more towards the scale of engineering collection and synthesis than proposing new physical fidelity or statistical generation mechanisms in simulation methods.**
>
> We acknowledge that the construction of SDTD is based on engineering-oriented approach. Its primary goal is to provide a large-scale and challenging benchmark for space debris tracking. Moreover, our real data results demonstrate that this dataset can still effectively support model training and transfer in practical observation scenarios.
>
> We appreciate the reviewer’s suggestions, and we will explore such higher-fidelity simulation mechanisms in future extensions of SDTD. Thank you for your suggestion.
>
> **2. The overall architecture of SDT-Net largely follows the existing multi-object tracking paradigm: it centers on detection-regression-association, using heatmap regression endpoints, embedding matching, and offset prediction to achieve temporal correlation. The proposed "Region-of-Interest Feature Enhancement (RoIFE)" module and "offset module" are essentially lightweight integrations of existing feature enhancement and motion offset ideas, without introducing new mechanisms in the algorithm's principles or optimization objectives.**
>
> Our goal is to provide a simple and effective solution that is user-friendly for the community and meets the needs of real-time computation in astronomy. Nevertheless, We made simple task-specific attempts for the debris, such as reformulating the target as an endpoint-based line source, as our primary goal is to maintain high efficiency for real-time astronomical tracking.
>
> We will further design astronomy specific network components in future work to improve performance. Thank you for your suggestion.
>
> **3. Train/val clips are mostly 1–4 frames, whereas test sets include sequences up to 30 frames (dense set concentrated around ~18). It’s unclear whether training on very short clips biases the tracker or underutilizes temporal cues.**
>
> Thank you for your insightful suggestion. To address the concern that training on short clips might limit the model's ability to utilize temporal cues, we additionally generated a set of long-sequence debris videos for training, consisting of 2,500 videos with 30,500 frames (average length ~12 frames).
>
> To keep the overall training scale comparable to the original setting, we mixed these with 3,500 videos randomly sampled from the original training set. The combined dataset contains 6,000 videos with 38,760 frames, which is similar in size to the original training set (~37,829 frames).
>
> After retraining on this new dataset, the results are as follows:
>
> | -                      | MOTA | HOTA | DetA  | AssA |
> | ---------------------- | ---- | ---- | ---- | ---- |
> | Baseline (Short clips) | 70.3 | 73.6 | 81.6 | 65.6 |
> | Retrained (Long clips) | 73.9 | 74.9 | 82.9 | 67.5 |
>
> We observed noticeable improvements in dense and long clips. We sincerely appreciate your suggestion and will include these results and discussions in the appendix.

---

> > ### Comment · Reviewer_y5Cr · 2025-11-28
> > **Thanks to the authors for their rebuttal**
> >
> > Thank you to the authors for their detailed rebuttal. Most of my questions have been resolved. Therefore, I maintain my original positive rating.

---

### Official Review · Reviewer_9kga · 2025-11-06

**Soundness:** 2
**Presentation:** 2
**Contribution:** 2
**Rating:** 4
**Confidence:** 3

**Summary:**

The paper presents SDT-Net, a deep learning framework for tracking space debris in complex skylight backgrounds, and introduces the Space Debris Tracking Dataset (SDTD), a large-scale synthetic dataset containing 18,040 video sequences with 62,562 frames and 250,000 synthetic debris instances. SDT-Net integrates feature enhancement, detection, and tracking modules to achieve high accuracy in cluttered, occluded, and dense debris environments. Evaluations on both synthetic and real Antarctic telescope data demonstrate strong performance, achieving a 73.2% MOTA score. The study highlights the potential of deep learning for real-time, transferable debris tracking and establishes SDTD as a benchmark for future research.Is this conversation helpful so far.

**Strengths:**

The paper proposes a deep-learning approach for space debris tracking in complex skylight backgrounds. The main contributions are:

1.The authors introduce a novel dataset, the Space Debris Tracking Dataset (SDTD), created by an observation-based simulation scheme, drawing on astronomy images (from e.g. the Zwicky Transient Facility, ZTF) and synthetically imposing debris trajectories and backgrounds. The dataset reportedly includes 18,040 video sequences (≈ 62,562 frames) and ~250,000 synthetic debris instances.
Moonlight

2.They propose a network named SDT‑Net, which comprises a Region-of-Interest Feature Enhancement (RoIFE) module, a detection module and a tracking module (tracking by detection plus association across frames). The network is targeted at the multi-object tracking (MOT) task in astronomical / debris scenarios.

3.They conduct experiments on their synthetic dataset and also evaluate transfer to real-world data: they claim a MOTA score (Multiple Object Tracking Accuracy) of ~73.2% (or ~70.6% in some versions) on a small real dataset collected at an Antarctic station.

4.They argue that their dataset addresses the paucity of annotated debris-tracking data, and that SDT-Net exhibits robustness under dense debris, occlusion, and complex star-field backgrounds.

Thus the paper is an attempt to bring modern deep-MOT methods into the space-debris tracking domain, supported by a large synthetic benchmark.

**Weaknesses:**

While the paper makes interesting advances, there are several concerns and weaknesses that the authors should address:

1. Synthetic-to-real transfer gap / dataset realism

1) Although the dataset is large and simulation-based, synthetic data may not fully replicate the statistical characteristics of real debris tracks, noise sources, background clutter, telescope artefacts, or imaging conditions (e.g., atmospheric scintillation, streak brightness variation, non-uniform PSF, variations in exposure times, sensor noise). The authors do test on a small real dataset, but the size is tiny (36 video sequences, ~2,228 frames) and limited to one station (Antarctic). This raises questions about generalisability to other sensors, orbital regimes, debris sizes, lighting conditions, star-field densities.

2) The paper reports a single performance number on real data; more extensive evaluation across different observational setups would strengthen the claim of “strong transferability”.

2. Dataset annotation / ground-truth fidelity and bias

1) The synthetic generation process may introduce biases (e.g., debris speed, size, appearance, background variation) that favour their method, especially since the method is trained on the synthetic data. It’s unclear how well annotation errors, occlusion patterns, sensor artefacts, and false positives/negatives are handled.

2) The real data annotations (astronomy experts) are limited in quantity; the annotation criteria, inter-annotator consistency, debris definitions (what qualifies as debris vs star/artefact) may affect reproducibility.

3. Evaluation metrics and baseline comparisons

1) The paper uses MOTA as a key metric; however, MOTA alone may not capture fine issues like ID-switches, fragmentation, false alarms in cluttered star fields, long-term track survival, or tracking latency (important for real-time/operational use).

2) The baselines compared are relatively generic MOT methods (e.g., CenterTrack, OCSORT) rather than domain-specific methods tailored to astronomical debris or long-exposure streak detection. A stronger argument would include recent astronomy/space-debris tracking methods.

3) The paper claims “state-of-the-art”, but many details about run-time, sensor input frame rate, false positive/false negative rates, resource usage (GPU/CPU) are missing. For an operational system, these are important.

4. Scalability and real-time viability

1) Space debris tracking in real operational settings often demands real-time or near-real-time performance, dealing with many debris objects, variable frame rates, large fields of view, and possibly resource-constrained platforms. The paper does not sufficiently discuss latency, computational load, or memory constraints.

2) Dense debris scenarios (e.g., mega-constellations, low Earth orbit clutter) may stress the method beyond the distribution of synthetic data; how well does it scale beyond the densities in the dataset?

5. Lack of orbital/physical modelling integration

1) The method appears largely vision-based (image/video processing) without explicit incorporation of orbital dynamics, sensor geometry, debris kinematics, or space situational awareness (SSA) context (e.g., Two-Line Elements, orbital propagation). In many practical applications, combining image tracking with orbital dynamics yields more robust performance. The paper doesn’t show how their output could link to orbit prediction or catalogue maintenance.

2) Without physics-based constraints (motion models, known debris motion patterns), the tracker may fail in ambiguous scenarios (e.g., overlapping tracks, rapid acceleration, non-linear motion), and the paper does not explore these limitations in depth.

6. Generalisation to other observational platforms

1) The dataset is constructed from ZTF images (ground-based optical telescope) and the real evaluation is from a single station. It is unclear how well the method would generalise to different sensors: e.g., space-based optical imagers, radar, different exposure times, spectral bands, or telescopes with different PSFs, different background noise levels, different orbital altitudes.

2) The authors should discuss how the method would adapt to e.g., GEO, MEO, or LEO regimes, or to different sensors (infrared, radar) or daytime/nighttime imaging.

7. Benchmark release and reproducibility

1) The paper mentions that “dataset and code will be released soon”. Without immediate availability, reproducibility and community uptake may be limited. The authors should commit to making the dataset, annotations, evaluation scripts and code available under a clear license, and provide a leaderboard or standard evaluation split.

2) If synthetic only, there is a risk that future users will duplicate their simulation bias. Clear documentation of simulation parameters, debris motion models, background modelling is needed.

8. Limited real-world deployment discussion

1) The paper could benefit from a deeper discussion of how this tracking method would integrate into operational debris tracking pipelines, what the false alarm risk is, how track continuity and object correlation across multiple passes/sensors would be handled, and what the end-to-end system implications are (e.g., collision avoidance, catalogue updating).

2) It is also unclear how many frames per second, what field of view, and what detection sensitivity (size/magnitude of debris) the system supports; practical relevance to e.g., <10 cm debris tracking is not characterised.

**Questions:**

Here are several relevant prior works that the authors should cite, covering space-debris detection/tracking, multi-object tracking in astronomy, datasets, and physics-informed approaches:

**Space-debris detection/tracking and optical observations**

Cament, L. et al., “Space Debris Tracking with the Poisson Labeled Multi-Bernoulli Multi-target Tracking Filter”, Sensors, 21(11):3684, 2021.

“A Robust Vision-based Algorithm for Detecting and Classifying Small Orbital Debris” (NASA MSFC) – algorithm for small debris using optical detection.
NASA Technical Reports Server

Navya, M. et al., “Deep Learning-Based Space Debris Tracking and Mitigation”, J Electrical Systems, 20(1):606-611, 2024.

Zhou, D., Sun, G., Zhang, Z., Wu, L., “On Deep Recurrent Reinforcement Learning for Active Visual Tracking of Space Non-cooperative Objects”, arXiv:2212.14304, Dec 2022.

Roll, D. S., Kurt, Z., Woo, W. L., “CosmosDSR – a methodology for automated detection and tracking of orbital debris using the Unscented Kalman Filter”, arXiv:2310.17158, Oct 2023.

**Astronomical multi-object tracking / star-field object tracking**

Guan, J., Cheng, H-Y., Wu, Y-P., Tian, C., Qi, J-Y., “Multi-target tracking for star sensor based on CenterTrack deep learning model”, Scientific Reports 15:37125 (2025).

**Space-debris modelling / simulation and environment context**

Kim, et al., “Review of Space Debris Modeling Methods and Development Trends”, Journal of Astronautical Sciences, 41(4):209-… (2024)

ESA Space Debris Environment Report, https://sdup.esoc.esa.int/discosweb/statistics/, sdup.esoc.esa.int

**Deep learning object/tracking methods in cluttered/low SNR astronomical settings**

SDebrisNet: “SDebrisNet: A Spatial–Temporal Saliency Network for Space Debris”, Applied Sciences 13(8):4955 (2023).

**Benchmarks/datasets for debris/satellite detection**

The authors should mention existing optical/space-object detection datasets, even if only for detection (not tracking) to position their contribution. For example, the Kaggle “Debris Detection Dataset” (optical images) – though limited.

**Orbit/dynamics embedding into tracking**

Although not directly DL-tracking, works that link vision tracking with orbit dynamics may strengthen the discussion. For example the PINN-based tracking after collision: “Tracking an Untracked Space Debris After an Inelastic Collision Using Physics Informed Neural Network”, arXiv:2307.09938 (2023).

---

> ### Author Response · Authors · 2025-11-21
> **Reply to Weaknesses 1**
>
> Thanks for your valuable feedback and efforts.
>
> **1.1-1. Although the dataset is large and simulation-based, synthetic data may not fully replicate the statistical characteristics of real debris tracks, noise sources, background clutter, telescope artefacts, or imaging conditions (e.g., atmospheric scintillation, streak brightness variation, non-uniform PSF, variations in exposure times, sensor noise).**
>
> Our simulation data have covered most of the scenarios of you have mentioned. As shown in Appendix Table 6: (1) Noise sources: adding different noise to debris. (2) streak brightness variation: simulating the uneven brightness of real debris under long exposure. (3) non-uniform PSF: width variation and diffusion standard deviation $δ_{psf}$, used to reproduce the non-uniform PSF of real telescopes. (4) variations in exposure times: since debris length reflects exposure time, sampling a wide length range encodes diverse exposure and motion conditions. These items (1)–(4) correspond respectively to the Noise, PSF(S), PSF parameter set, and Length configurations in Table 6. In addition, the original ZTF image contains a large amount of background clutter, telescope artefacts, atmospheric scintillation and sensor noise (see Fig. 6, Appendix B.1).
>
> **1.1-2. The authors do test on a small real dataset, but the size is tiny (36 video sequences, ~2,228 frames) and limited to one station (Antarctic). This raises questions about generalisability to other sensors, orbital regimes, debris sizes, lighting conditions, star-field densities.**
>
> Our real data are sufficiently diverse to serve as a reliable and fair evaluation set. Because (1) our real-world observations already cover a large portion of the accessible sky, as all fields are taken above 30° altitude, the data naturally include a wide range of real-world conditions, such as varying skylight backgrounds, atmospheric noise levels, debris brightness, streak lengths, and non-uniform PSFs. (2) The evaluation is not affected by the dataset size, as there is an inherent domain gap between synthetic and real debris. The results further demonstrate that a model trained entirely on large-scale synthetic data can still generalize robustly to unseen real scenes.
>
> Our model is generalizable in other sensors, orbital regimes, debris sizes, lighting conditions, star-field densities. The reasons are as follows: (1) In terms of sensor variation, the simulated data and the real data originate from different sensors, yet the model still achieves stable performance on the real data, indicating robustness to sensor changes. (2) The changes in debris velocity, tail length and brightness reflected by the orbital state have been fully covered in our simulation settings. (3) The diversity of debris size (including length and width) is also extensively modeled in the synthetic data. (4) We have included a variety of lighting conditions in the simulation to cover more imaging differences in real scenes. (5) The ZTF background used in the synthesis itself contains a complex and variable star structure, including different star field densities. The relevant settings in points (2) to (4) above all correspond directly to the parameter configurations listed in Table 6.
>
> **1.2. The paper reports a single performance number on real data; more extensive evaluation across different observational setups would strengthen the claim of “strong transferability”.**
>
> Thank you for your suggestion. To further evaluate transferability under different observational setups, we synthesized three additional datasets using backgrounds from HST, SDSS, and ZTF. We directly applied the pre-trained SDT-Net for inference. The results are as follows:
>
> | SDT-Net       | MOTA | HOTA | Det  | AssA |
> | ------------- | ---- | ---- | ---- | ---- |
> | HST(F814W)    | 66.7 | 64.5 | 67.2 | 61.9 |
> | SDSS(r-band)  | 67.4 | 69.4 | 71.9 | 66.9 |
> | ZTF(g-band)   | 69.7 | 72.1 | 74.9 | 69.5 |
> | SDTD(dataset) | 70.3 | 73.6 | 81.6 | 65.6 |
>
> These results not only demonstrate the effectiveness of SDT-Net, but its strong cross-domain performance further confirms that the real-data evaluation is reliable.

---

> ### Author Response · Authors · 2025-11-21
> **Reply to Weaknesses 2 to 3.2**
>
> **2.1-1. The synthetic generation process may introduce biases (e.g., debris speed, size, appearance, background variation) that favour their method, especially since the method is trained on the synthetic data.**
>
> Our dataset does not exhibit significant bias, due to it covers a sufficiently large diversity. All simulation parameters are randomly sampled to ensure broad variation, and according to our astronomy experts, these parameter ranges already cover most debris conditions observed in practice. Table 6 (Appendix) lists the full parameter ranges for debris speed (displacement), size (length/width), appearance (length/width, PSF, fracture patterns, noise), and background variation (complex ZTF skylight conditions) used in SDTD.
>
> **2.1-2. It’s unclear how well annotation errors, occlusion patterns, sensor artefacts, and false positives/negatives are handled.**
>
> Our dataset does not contain annotation errors, as all debris annotations are accurate, directly extracted from simulation.
>
> Except that, SDTD explicitly includes occlusion patterns. By generating debris with randomized multi-object trajectories, we naturally obtain spatial and temporal intersections. Also, the dataset contains a variety of sensor artefacts（such as observational noise, optical PSF distortions, slight telescope jitter, and electronic striping). These are illustrated in Appendix Fig. 6.
>
> **2.2. The real data annotations (astronomy experts) are limited in quantity; the annotation criteria, inter-annotator consistency, debris definitions (what qualifies as debris vs star/artefact) may affect reproducibility.**
>
> The annotation quality of the real data is reliable, as the object tracking annotations are clearly defined as pixel locations. The annotation protocol follows the same principles as COCO-style labeling, usually not requiring additional annotation criteria or inter-annotator inconsistencies. The annotated dataset will be publicly released after acceptance for community check.
>
> Concerning the definition of debris, we have clearly stated in Section 3 (first paragraph) that “debris often appears as a line source in the images.” In the real data, debris is defined strictly by line source characteristics, whereas stars appear as point sources. This distinction is unambiguous and fully consistent with astronomical conventions, and thus does not introduce ambiguity or impact reproducibility.
>
> **3.1. The paper uses MOTA as a key metric; however, MOTA alone may not capture fine issues like ID-switches, fragmentation, false alarms in cluttered star fields, long-term track survival, or tracking latency (important for real-time/operational use).**
>
> Our key metric is not just MOTA, as shown in Table 1 of the original submission. We have included several key metrics: (1) IDF1 and IDS measure identity consistency. They explicitly quantify ID switches, and fragmentation-related inconsistencies are implicitly captured through the identity-matching process.(2) HOTA provides a much stronger characterization of **long-term tracking** behavior. By jointly evaluating detection accuracy and association accuracy at every frame, and by integrating these factors across the entire sequence, HOTA directly measures a tracker’s ability to maintain consistent identities over extended time spans. (3) DetA and AssA for detection precision and association performance. DetA is directly affected by false positives and therefore reflects **false alarm** behavior in **cluttered backgrounds**. Together, these metrics provide a comprehensive evaluation of the tracking task, rather than relying on MOTA alone.
>
> Appendix Table 7 shows that SDT-Net achieves an inference time of 0.34 s per image, indicating its operational latency is suitable for real-time use.
>
> **3.2. The baselines compared are relatively generic MOT methods (e.g., CenterTrack, OCSORT) rather than domain-specific methods tailored to astronomical debris or long-exposure streak detection. A stronger argument would include recent astronomy/space-debris tracking methods.**
>
> There is no released domain-specific debris **tracking** for comparison because existing methods are almost debris **detection** rather than tracking. They cannot tackle dense objects and also cannot treat association. So, we choose the vision multi-object tracking methods for comparison.
>
> The generic MOT methods (e.g., CenterTrack, OCSORT) used in the current evaluation have been fully trained on SDTD, providing comparable tracking outputs. Thus, we believe that they could represent the feasible and fair baselines for this newly defined task.

---

> > ### Author Response · Authors · 2025-11-21
> > **Reply to Weaknesses 3.3 to 5.1**
> >
> > **3.3. The paper claims “state-of-the-art”, but many details about run-time, sensor input frame rate, false positive/false negative rates, resource usage (GPU/CPU) are missing. For an operational system, these are important.**
> >
> > Runtime has been provided at Appendix Table 7 of the original submission, showing 0.34s per image. Frame-rate statistics have been shown in Appendix B.5.1 and Fig. 9 of the original submission.
> >
> > For explicitly showing false positives and false negatives, we thank you for your suggestion and have included the corresponding results and will update them in the rebuttal revision:
> > | - | FP | FN |
> > | -------- | -------- | -------- |
> > | debris test     | 333     | 596     |
> > | Dense debris test     | 1698    |  1928   |
> >
> > The hardware setup could be found in Section 5.1 and Appendix Table 5 of the original submission, and we will include the Intel Xeon Platinum 8558 CPU detail in the revision.
> >
> >
> > **4.1.Space debris tracking in real operational settings often demands real-time or near-real-time performance, dealing with many debris objects, variable frame rates, large fields of view, and possibly resource-constrained platforms. The paper does not sufficiently discuss latency, computational load, or memory constraints.**
> >
> > The runtime and resource-related results have been provided in Section 5.1 and Appendix Tables 5 and 6 of the original submission, showing SDT-Net achieves real-time performance with modest computational resources. In particular, on a single RTX 4090 GPU, SDT-Net processes one 1524×1524 astronomical image in 0.34s. This represents a substantial improvement over traditional astronomical debris detection pipelines. Such as the StreakDet requires 13s per 2k×2k image.
> >
> > **4.2. Dense debris scenarios (e.g., mega-constellations, low Earth orbit clutter) may stress the method beyond the distribution of synthetic data; how well does it scale beyond the densities in the dataset?**
> >
> > Our evaluation on real-world data demonstrates the method’s robustness in dense debris scenarios. (1) We already included dense debris scenarios in our synthetic data generation, so such cases are not out of the distribution of our synthetic dataset. (2) Our real data contains dense debris scenarios, and our method achieves good performance under these settings, which proves its generalization ability.
> >
> > **5.1-1. The method appears largely vision-based (image/video processing) without explicit incorporation of orbital dynamics, sensor geometry, debris kinematics, or space situational awareness (SSA) context (e.g., Two-Line Elements, orbital propagation). In many practical applications, combining image tracking with orbital dynamics yields more robust performance.**
> >
> > We agree that this is an important research direction, and in future work we plan to incorporate orbital dynamics, sensor geometry, and debris motion models to build more complex and physically consistent simulations and methods. The primary goal of this work is to evaluate the capability of vision-based models to detect and track debris in astronomical scenarios using a simulated debris dataset. Our experimental results show that AI models can achieve stable and effective tracking performance.
> >
> > **5.1-2. The paper doesn’t show how their output could link to orbit prediction or catalogue maintenance.**
> > It is straightforward to integrate our system into existing astronomical processing pipelines. In the current debris processing pipeline, a debris-detection module should take a FITS (an image format in astronomy) image as input and output the sky coordinates of each detected object. Our model follows the same interface: it is trained on raw FITS-domain images, so it has the ability to directly operate FITS images. And its predicted pixel-level locations can be converted into standard sky coordinates (Right Ascension and Declination) using the WCS (World Coordinate System) metadata stored in every FITS file. This means the outputs of SDT-Net are immediately usable by downstream astronomy modules—such as orbit estimation, trajectory prediction, and catalogue updating—without any additional transformation or engineering effort. These information will be added in the revision.

---

> > > ### Author Response · Authors · 2025-11-21
> > > **Reply to Weaknesses 5.2 to 7.2**
> > >
> > > **5.2.Without physics-based constraints (motion models, known debris motion patterns), the tracker may fail in ambiguous scenarios (e.g., overlapping tracks, rapid acceleration, non-linear motion), and the paper does not explore these limitations in depth.**
> > >
> > > Although our simulation does not incorporate full physics-based constraints, the real-world evaluation shows that our vision-based formulation remains effective. Among the ambiguous scenarios mentioned, some are already covered in SDTD, while others do not occur in our observational setting.
> > >
> > > Although our simulation does not incorporate full physics-based constraints, our real-world experiments show that the vision-based tracking formulation remains effective in practice. The ambiguous scenarios raised by the reviewer can be divided into two categories: those already covered in our dataset, and those that do not occur in our observational setting.
> > >
> > > For the scenarios we do cover, overlapping or intersecting tracks are already modeled in SDTD. As noted in our response to Comment 2.1, the randomized multi-object trajectories naturally create spatial and temporal intersections, leading to realistic occlusions and partial overlaps.
> > >
> > > For the scenarios that do not occur, rapid acceleration and strongly non-linear motion are not present within the field of view of a single long-exposure astronomical telescope. Under typical exposure durations and survey configurations, debris motion is close to linear with nearly constant velocity, placing such cases outside the scope of our current observational regime.
> > >
> > > These factors will be considered as we extend our work to tracking artificial satellites in the future, and we appreciate the reviewer’s suggestion for future directions.
> > >
> > >
> > > **6.1. The dataset is constructed from ZTF images (ground-based optical telescope) and the real evaluation is from a single station. It is unclear how well the method would generalise to different sensors: e.g., space-based optical imagers, radar, different exposure times, spectral bands, or telescopes with different PSFs, different background noise levels, different orbital altitudes.**
> > >
> > > Thank you for your suggestion. As shown in our response to Comment 2, we have further provided results on other sensors, and we will update it in the Rebuttal Revision.
> > >
> > > To demonstrate the generalizability of our method across different sensors, we clarify the following points: (1) For space-based optical imagers and spectral bands, we compared SDT-Net on HST and ZTF g-band data in Comment 2. (2) Radar is not an instrument used in optical debris-tracking pipelines and therefore falls outside the scope of this work. (3) Different exposure times, PSFs, and background noise levels are explicitly simulated in our dataset (Table 6: Length, PSF parameters, Noise). Differences in orbital altitude are reflected through changes in debris speed, brightness, and length, all of which are also covered in the simulation ranges of Table 6.
> > >
> > > **6.2. The authors should discuss how the method would adapt to e.g., GEO, MEO, or LEO regimes, or to different sensors (infrared, radar) or daytime/nighttime imaging.**
> > >
> > > Our model can generalize to different orbital regimes because the training simulation already encompasses the variations associated with GEO, MEO, and LEO conditions. As noted in our response to Comment 6.1, differences across orbital regimes primarily manifest as changes in debris speed, apparent brightness, and streak length. These variations are explicitly covered in the parameter ranges of our dataset (Table 6).
> > >
> > > Infrared and radar sensors are almost not used for detecting debris. Similarly, daytime imaging is not applicable in astronomical debris observations, as the strong skylight background prevents effective observations. Hence, our method is designed for and evaluated under nighttime imaging.
> > >
> > > **7.1. The paper mentions that “dataset and code will be released soon”. Without immediate availability, ....., evaluation scripts and code available under a clear license, and provide a leaderboard or standard evaluation split.**
> > >
> > > Thank you for your suggestion. We commit that, once the paper is accepted, we will release the code, dataset, and the debris generation tool to ensure reproducibility and to benefit the community.
> > >
> > > **7.2. If synthetic only, there is a risk that future users will duplicate their simulation bias. Clear documentation of simulation parameters, debris motion models, background modelling is needed.**
> > >
> > > More details related to simulation bias are discussed in our response to Comment 2.1. Our experiments and the additional evaluations provided above demonstrate that the simulation does not introduce bias. All simulation parameters, debris motion models, and background modelling details are provide in Table 6. Moreover, the simulation tools will be publicly released, allowing users to check, modify, and extend the generation process as needed.

---

> > > > ### Author Response · Authors · 2025-11-21
> > > > **Reply to Weaknesses 8.1 to Questions**
> > > >
> > > > **8.1-1. The paper could benefit from a deeper discussion of how this tracking method would integrate into operational debris tracking pipelines**
> > > >
> > > > Thanks for your suggestion, about 'how this tracking method would integrate into operational debris tracking pipelines', we will added the contents of 5.1-2 in the final revision.
> > > >
> > > > **8.1-2. What the false alarm risk is**
> > > >
> > > > For false alarms risk, a quantitative assessment of false alarms is already provided in Table 1 through MOTA, ID-switches, and related metrics. On the qualitative side, false alarms typically arise from low-SNR or astronomical sensor artefacts that generate spurious line-like fake sources. We will explore more complex models in future work to further reduce false positives.
> > > >
> > > > **8.1-3. How track continuity and object correlation across multiple passes/sensors would be handled**
> > > >
> > > > For multi-pass or multi-sensor settings, the debris image coordinates predicted by SDT-Net can be directly transformed into the sky coordinates. Since debris does not change its orbit during flight, its position in other sensors can be easily predicted, making accurate cross-sensor matching possible, which is well studied in the astronomy and space science community.
> > > >
> > > > **8.1-4. What the end-to-end system implications are (e.g., collision avoidance, catalogue updating).**
> > > >
> > > > A more precise end-to-end tracking system has clear operational implications: (1) stable and continuous tracking support collision avoidance for spacecraft, (2) detecting and tracking newly observed debris assists with catalogue updating and maintenance, (3) high-effectiveness tracking improves the overall efficiency of debris monitoring workflows.
> > > >
> > > > **8.2. It is also unclear how many frames per second, what field of view, and what detection sensitivity (size/magnitude of debris) the system supports; practical relevance to e.g., <10 cm debris tracking is not characterised.**
> > > >
> > > > We report the frame distribution of the SDTD in Appendix Fig. 6. The field of view (FOV) is approximately 47 deg², which is a typical wide-field scale for ground-based telescopes. The detection sensitivity, including the full range of debris size and brightness parameters, is provided in Table 6.
> > > > For “<10 cm debris”, such objects are far below the detectability limit of ground-based optical imaging. Current optical systems typically detect debris (e.g., in the GEO region) at sizes of roughly ≥1 m (Nakajima et al., 2006).
> > > >
> > > > Nakajima A, Kurosaki H, Fukaya T. Space Debris Optical Observation System in JAXA/IAT[C]//Proceedings of the Advanced Maui Optical and Space Surveillance Technologies Conference. 2006: 10-14.
> > > >
> > > >
> > > > **Questions:
> > > >
> > > > Here are several relevant prior works that the authors should cite, covering space-debris detection/tracking, multi-object tracking in astronomy, datasets, and physics-informed approaches:**
> > > >
> > > > We thank the reviewer for the helpful suggestion. We will incorporate these related studies into the Related Work section to provide a more comprehensive coverage of prior research.

---

### Comment · Area_Chair_1qfV · 2025-11-26

Dear Reviewers

If you haven't done so, please engage with authors and respond to the authors who have made significant efforts in addressing your comments.

Please check if they addressed your comments or not, or to what degree.

Thanks

AC

---

### Meta-Review · Area_Chair_mQVu · 2026-01-10

**Summary:**

This paper receives the scores of 6, 6, 4, 4 (average 5.0). The main concerns raised by the reviewers include moderate novelty (components adapted from existing MOT); synthetic-to-real realism and breadth of real validation are limited (single station, small set); limited initial baselines (transformer trackers, recent SORT variants), unclear scalability/runtime reporting; and absence of physics/orbital dynamics integration or multi-sensor fusion.

**Reviewer Concerns:**

In the feedback, the authors added comparisons to MOTR, TrackFormer, FairMOT, PD-/Hybrid-SORT (all retrained on SDTD), cross-sensor synthetic tests (HST/SDSS/ZTF), longer-clip training that improves dense/long sequences, FP/FN counts, hardware details, loss-weight and backbone sensitivity analyses, and clarification on WCS integration and release plans.

These responses substantially improve baseline coverage and clarity but only partially address core concerns about physical fidelity and broader real-world validation.

**Reviewer Scores:**

Based on the reviewers' comments, it is reasonable to expect that the two reviewers who gave positive scores will maintain their evaluation, while the two reviewers who gave negative scores will also likely keep their scores unchanged, given that their concerns have not been fully addressed.

---

### Decision · Program_Chairs · 2026-01-26

Reject